



# Modelling spatial and temporal dynamics of GPP in the Sahel from earth observation based photosynthetic capacity and quantum efficiency

Torbern Tagesson[1], Jonas Ardö[2], Bernard Cappelaere[3], Laurent Kergoat[4], Abdulhakim M. Abdi[2], Stéphanie Horion[1], Rasmus Fensholt[1]

[1]Department of Geosciences and Natural Resource Management, University of Copenhagen, Øster Voldgade 10, DK-1350 Copenhagen, Denmark; E-Mails: torbern.tagesson@geo.ku.dk, stephanie.horion@geo.ku.dk, rf@geo.ku.dk

[2]Department of Physical Geography and Ecosystem Science, Lund University, Sölvegatan 12, SE- 223 62 Lund, Sweden, E-Mails: jonas.ardo@nateko.lu.se, hakim.abdi@gmail.com

[3]HydroSciences Montpellier, IRD, CNRS, Univ. Montpellier, Montpellier, France, E-Mail: bernard.cappelaere@um2.fr

[4]Geoscience Environnement Toulouse, (CNRS/UPS/IRD), 14 av E Belin, 31400 Toulouse, France, E-Mail: laurent.kergoat@get.obs-mip.fr

*Correspondence to*: Torbern Tagesson (torbern.tagesson@ign.ku.dk)

**Abstract.** It has been shown that vegetation growth in semi-arid regions is an important sink for human induced fossil fuel emissions of $CO_2$, which indicates the strong need for improved understanding, and spatially explicit estimates of $CO_2$ uptake (gross primary productivity (GPP)) in semi-arid ecosystems. This study has three aims: 1) to evaluate the MOD17A2H GPP (collection 6) product against eddy covariance (EC) based GPP for six sites across the Sahel. 2) To find evidence on the relationships between spatial and temporal variability in EC based photosynthetic capacity ($F_{opt}$) and quantum efficiency ($\alpha$) and earth observation (EO) based vegetation indices 3) To study the applicability of EO up-scaled $F_{opt}$ and $\alpha$ for GPP modelling purposes. MOD17A2H GPP (collection 6) underestimated GPP strongly, most likely because the maximum light use efficiency is set too low for semi-arid ecosystems in the MODIS algorithm. The intra-annual dynamics in $F_{opt}$ was closely related to the shortwave infrared water stress index (SIWSI) closely coupled to equivalent water thickness, whereas $\alpha$ was closely related to the renormalized difference vegetation index (RDVI) affected by chlorophyll abundance. Spatial and inter-annual dynamics in $F_{opt}$ and $\alpha$ were closely coupled to the normalized difference vegetation index (NDVI) and RDVI, respectively. Modelled GPP based on $F_{opt}$ and $\alpha$ up-scaled using EO based indices reproduced in situ GPP well for all but a cropped site. The cropped site was strongly impacted by intensive anthropogenic land use. This study indicates the strong applicability of EO as a tool for parameterising spatially explicit estimates of photosynthetic capacity and efficiency; incorporating this into dynamic global vegetation models could improve global estimations of vegetation productivity, ecosystem processes and biochemical and hydrological cycles.



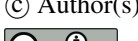

**Keywords:** Remote sensing, Gross Primary Productivity, MOD17A2H, light use efficiency, photosynthetic capacity,
quantum efficiency

## 1 Introduction

Vegetation growth in semi-arid regions is an important sink for human induced fossil fuel emissions. Semi-arid regions
are even the main biome driving long-term trends and inter-annual variability in carbon dioxide ($CO_2$) uptake by
terrestrial ecosystems (Ahlström et al., 2015; Poulter et al., 2014). It is thus important to understand the long-term
variability of vegetation growth in semi-arid areas and their response to environmental conditions to better quantify and
forecast the effects of climate change.
The Sahel is a semi-arid transition zone between the dry Sahara desert in the North and the humid Sudanian savanna
in the south. The region has experienced numerous severe droughts during the last decades that resulted in region-wide
famines in 1972-1973 and 1984–1985 and localized food shortages across the region in 1990, 2002, 2004, 2011 and
2012 (Abdi et al., 2014; United Nations, 2013). Vegetation productivity is thereby an important ecosystem service for
the people living in the Sahel, but it is under high pressure. The region experiences a strong population growth,
increasing the demand on the ecosystem services due to cropland expansion, increased pasture stocking rates and
fuelwood extraction (Abdi et al., 2014). Continuous cropping is practised to meet the demand of the growing population
and has resulted in reduced soil fertility, which affects vegetation productivity negatively (Samaké et al., 2005; Chianu
et al., 2006).
At the same time as we have reports of declining vegetation productivity, we have contradicting reports of greening of
the Sahel based on remote sensing data (Dardel et al., 2014; Fensholt et al., 2013). The greening of the Sahel has mainly
been attributed to alleviated drought stress conditions due to increased precipitation since the mid-1990s (Hickler et al.,
2005). Climate is thus another important factor regulating vegetation productivity and semi-arid regions, such as the
Sahel, are particularly vulnerable to climate fluctuations due to their vulnerability to moisture conditions.
Estimation of gross primary productivity (GPP), i.e. uptake of atmospheric $CO_2$ by vegetation, is still a major
challenge within remote sensing of ecosystem services. GPP is a main driver of ecosystem services such as climate
regulation, carbon (C) sequestration, C storage, food production, or livestock grassland production. Within earth
observation (EO), spatial quantification of GPP generally involves light use efficiency (LUE), defined as the efficiency
to convert absorbed solar light into $CO_2$ uptake (Monteith, 1972, 1977). It has been shown that LUE varies in space and
time due to factors such as plant functional type, drought and temperature, nutrient levels and physiological limitations
of photosynthesis (Garbulsky et al., 2010; Paruelo et al., 2004; Kergoat et al., 2008). The LUE concept has been applied
using various methods, either by using a biome-specific LUE constant (Ruimy et al., 1994), or by modifying a
maximum LUE using meteorological variables (Running et al., 2004).
An example of an LUE based model is the standard GPP product from the Moderate Resolution Imaging
Spectroradiometer (MODIS) sensor (MOD17A2). Within the model, absorbed photosynthetically active radiation
(PAR) is estimated as a product of the fraction of PAR absorbed by the green vegetation (FPAR from MOD15A2)
multiplied with daily PAR from the meteorological data of the Global Modeling and Assimilation Office (GMAO). A
set of maximum LUE parameters specified for each biome are extracted from a Biome Properties Look-Up Table
(BPLUT). Then maximum LUE is modified depending on air temperature ($T_{air}$) and vapor pressure deficit (VPD) levels





(Running et al., 2004). Sjöström et al. (2013) evaluated the MOD17A2 product (collection 5.1) for Africa, and showed
that it was underestimating GPP for semi-arid savannas in the Sahel. Explanations for this underestimation were that the
assigned maximum LUE from the BPLUT is set too low and uncertainties in the FPAR (MOD15A2) product. Recently,
a new collection of MOD17A2 at 500 m spatial resolution was released (MOD17A2H; collection 6) with an updated
BPLUT, updated GMAO meteorological data,  improved quality control and gap filling of the FPAR data from
MOD15A2 (Running and Zhao, 2015).
It has been shown that the LUE method does not perform well in arid conditions and at agricultural sites (Turner et
al., 2005). Additionally, the linearity assumed by the LUE model is usually not found  as the response of GPP to
incoming light follows more of an asymptotic curve (Cannell and Thornley, 1998). Investigating other methods for
remotely determining GPP is thus of great importance, especially for semi-arid environments. Therefore, instead of
LUE we focus on the light response function of GPP at the canopy scale, and spatial and temporal variation of its two
main parameters: maximum GPP under light saturation (canopy-scale photosynthetic capacity; $F_{opt}$), and the initial
slope of the light response function (canopy-scale quantum efficiency; α) (Falge et al., 2001; Tagesson et al., 2015a).
Photosynthetic capacity is a measure of the maximum rate at which the canopy can fix $CO_2$ during photosynthesis
($\mu$mol $CO_2$ $m^{-2}$ $s^{-1}$) whereas α is the amount of $CO_2$ fixed per incoming PAR ($\mu$mol $CO_2$ $\mu$mol $PAR^{-1}$). Just to clarify the
difference in LUE and α in this study; LUE ($\mu$mol $CO_2$ $\mu$mol $APAR^{-1}$) is the slope of a linear fit between $CO_2$ uptake
and absorbed PAR, whereas α ($\mu$mol $CO_2$ $\mu$mol $PAR^{-1}$) is the initial slope of an asymptotic curve against incoming
PAR.
It has been proven that $F_{opt}$ and α are closely related to chlorophyll abundance due to their coupling with the electron
transport rate (Ide et al., 2010). Additionally, in semi-arid ecosystems water availability is generally considered to be
the main limiting factor affecting intra-annual dynamics of vegetation growth (Fensholt et al., 2013; Hickler et al.,
2005; Tagesson et al., 2015b). Several remote sensing studies have established relationships between remotely sensed
vegetation indices and ecosystem properties such as chlorophyll abundance and equivalent water thickness (Yoder and
Pettigrew-Crosby, 1995; Fensholt and Sandholt, 2003). In this study we will analyse if EO vegetation indices can be
used for up-scaling $F_{opt}$ and α and investigate if this could offer a promising way to map GPP in semi-arid areas. This
potential will be analysed by the use of detailed ground observations from six different measurement sites (eddy
covariance flux towers) across the Sahel.
The three aims of this study are:

1) To evaluate the recently released MOD17A2H GPP (collection 6) product and to investigate if it is better at

capturing GPP levels for the Sahel than collection 5.1. We hypothesise that MOD17A2H GPP (collection 6)

product will estimate GPP well for the six Sahelian measurement sites, because of  the major changes done in

comparison to collection 5.1  (Running and Zhao, 2015).

2) To find evidence on the relationships between spatial and temporal variability in $F_{opt}$ and α and remotely

sensed vegetation indices. We hypothesise that remotely sensed vegetation indices that are closely related to

chlorophyll abundance can be used for quantifying spatial and inter-annual dynamics in $F_{opt}$ and α. Vegetation

indices closely related to equivalent water thickness are closely linked to intra-annual dynamics in $F_{opt}$ and α

across the Sahel.

3) To evaluate the applicability of a GPP model based on the light response function using remotely sensed

vegetation indices and incoming PAR as input data.





**2 Materials and Methods**
**2.1 Site description**
The Sahel stretches from the Atlantic Ocean in the west to the Red Sea in the east. The northern border towards the
Sahara and the southern border towards the humid Sudanian Savanna are defined by the 150 and 700 mm isohytes,
respectively (Fig. 1) (Prince et al., 1995). Tree and shrub canopy cover is now generally low (< 5%) and dominated by
species of *Balanites*, *Acacia*, *Boscia* and *Combretaceae* (Rietkerk et al., 1996). Annual grasses such as *Schoenefeldia*
*gracilis*, *Dactyloctenium aegypticum*, *Aristida mutabilis*, and *Cenchrus biflorus* dominate the herbaceous layer, but
perennial grasses such as *Andropogon gayanus*, *Cymbopogon schoenanthus* can also be found (Rietkerk et al., 1996; de
Ridder et al., 1982). From the FLUXNET database (Baldocchi et al., 2001), we selected the six available measurement
sites with eddy covariance based $CO_2$ flux data from the Sahel (Table 1; Fig. 1). The sites represent a variety of the
ecosystems present in the region, from dry fallow bush savanna to seasonally inundated acacia forest. For a full
description of the measurement sites, we refer to (Tagesson et al., 2016) and the references in Table 1.
<Table 1>
<Figure 1>

**2.2 Data collection**
**2.2.1 Eddy covariance, hydrological and meteorological in situ data**
Eddy covariance (EC), hydrological and meteorological data originating from the years between 2005 and 2013 were
collected from the principal investigators of the measurement sites (Tagesson et al., 2016). The EC sensor set-up
consisted of open-path $CO_2/H_2O$ infrared gas analysers and 3-axis sonic anemometers. Data were collected at 20 Hz rate
and statistics were calculated for 30-min periods. For a full description of sensor set up and post processing of the EC
data, see references in Table 1. Final fluxes were filtered according quality flags provided by FLUXNET and outliers
were filtered according Papale et al. (2006). We extracted the original net ecosystem exchange (NEE) data without any
gap-filling or partitioning of NEE to GPP and ecosystem respiration. We also collected hydrological and meteorological
data: air temperature ($T_{air}$; °C), rainfall (P; mm), relative air humidity (Rh; %), soil moisture at 0.1 m depth (SWC; %
volumetric water content), incoming global radiation ($R_g$; W m$^{-2}$), incoming photosynthetically active radiation (PAR;
μmol m$^{-2}$ s$^{-1}$), VPD (hPa), peak dry weight biomass (g dry weight m$^{-2}$), C3/C4 species ratio, and soil conditions
(nitrogen and C concentration; %). For a full description of the collected data and sensor set-up, see Tagesson et al.

(2016).


**2.2.2 Earth Observation data and gridded ancillary data**
Remotely sensed composite products from the MODIS/Terra L4 from the Sahel were collected at Reverb ECHO
(NASA, 2016). The collected products were GPP (MOD17A2H; collection 6) and the Nadir Bidirectional Reflectance
Distribution Function (BRDF) adjusted reflectance (NBAR) (8-day composites; MCD43A4; collection 5.1) at 500*500
m$^2$ spatial resolution, and the normalized difference vegetation index (NDVI), and the enhanced vegetation index (EVI)
(16-day composites; MOD13Q1; collection 6) at 250*250 m$^2$ spatial resolution. The NBAR product was preferred over
the reflectance product (MOD09A1), in order to avoid variability caused by varying sun and sensor viewing geometry
(Huber et al., 2014; Tagesson et al., 2015c). We extracted the median of the 3x3 pixels centred at the location of the EC
towers. The time series of the remotely sensed products were filtered according the MODIS quality control data;



MOD17A2H is a gap-filled and filtered product, QC data from MCD43A2 were used for the filtering of MCD43A4;
and bit 2-5 (highest –decreasing quality) was used for MOD13Q1. Finally, data were gap-filled to daily values using
linear interpolation.
For a GPP model to be applicable on a larger spatial scale, a gridded data set of incoming PAR is needed. We
downloaded ERA Interim reanalysis PAR at the ground surface (W m$^{-2}$) with a spatial resolution of 0.25°×0.25°
accumulated for each 3-hour period 2000-2015 from the European Centre for Medium-Range Weather Forecasts
(ECMWF) (Dee et al., 2011; ECMWF, 2016).

**2.3 Data handling**
**2.3.1 Intra-annual dynamics in photosynthetic capacity and quantum efficiency**
Both linear and hyperbolic equations have been used for investigating the response of GPP to incoming light (Wall and
Kanemasu, 1990; Campbell et al., 2001). However, they do not represent the lower part of the light response function
particularly well, and we thereby instead choose to use the asymptotic Mitscherlich light-response function (Inoue et al.,
2008; Falge et al., 2001). The Mitscherlich light-response function was fitted between daytime NEE and incoming
PAR:
$$NEE = -(F_{opt}) \times \left(1 - e^{\left(\frac{-\alpha \times PAR}{F_{opt}}\right)}\right) + R_d \qquad (1)$$
where $F_{opt}$ is the $CO_2$ uptake at light saturation (photosynthetic capacity; μmol $CO_2$ m$^{-2}$ s$^{-1}$), $R_d$ is dark respiration
(μmol $CO_2$ m$^{-2}$ s$^{-1}$), and α is the initial slope of the light response curve (quantum efficiency; μmol $CO_2$ μmol PAR$^{-1}$)
(Falge et al., 2001). By subtracting $R_d$ from Eq. 1, the function is forced through zero and GPP is thereby estimated. We
fitted Eq. 1 using 7-day moving windows with 1 day time steps and generating daily values of $F_{opt}$ and α. To assure high
quality of the fitted parameters, parameters were excluded from the analysis when the fitting was insignificant (p-
value<0.05), and when they were out of range ($F_{opt}$ and α >peak value of the rainy season times 1.2). Additionally,
outliers were filtered following the method by Papale et al. (2006) using 30-day moving windows with 1 day time steps.

**2.3.2 Vegetation indices**
We analysed the relationship between $F_{opt}$, α and some commonly applied vegetation indices:
The maximum absorption in the red wavelengths generally occurs at 682 nm as this is the peak absorption for
chlorophyll a and b (Thenkabail et al., 2000), which makes vegetation indices that include the red band sensitive to
chlorophyll abundance. By far the most common vegetation index is the NDVI (Rouse et al., 1974):
$$NDVI = \frac{(\rho_{NIR} - \rho_{red})}{(\rho_{NIR} + \rho_{red})} \qquad (2)$$
where $\rho_{NIR}$ is the reflectance factor in the near infrared (NIR) band (band 2) and $\rho_{red}$ is the reflectance factor in the red
band (band 1). The NIR radiance is scattered by the air-water interfaces between the cells whereas red radiance is
absorbed by chlorophyll and its accessory pigments (Gates et al., 1965). Normalization is done to reduce effects of
atmospheric errors, solar zenith angles, and sensor viewing geometry, as well as increasing the vegetation signal (Qi et
al., 1994; Inoue et al., 2008).





A well-known issue with the NDVI is that it saturates at high biomass because the absorption of red light at ~670 nm
peaks at higher biomass loads whereas NIR reflectance continues to increase due to multiple scattering effects
(Mutanga and Skidmore, 2004; Jin and Eklundh, 2014). By reducing atmospheric and soil background influences, EVI
increases the signal from the vegetation and maintain sensitivity in high biomass regions (Huete et al., 2002).
$$EVI = G \frac{(\rho_{NIR} - \rho_{red})}{(\rho_{NIR} + C_1\rho_{red} - C_2\rho_{blue} + L)} \qquad (3)$$
where $\rho_{blue}$ is the reflectance factor in the blue band (band 3). The coefficients $C_1=6$ and $C_2=7.5$ correct for atmospheric
influences, while $L=1$ adjust for the canopy background. The factor $G=2.5$ is the gain factor.
Another attempt to overcome the issue of NDVI saturation was proposed by Roujean and Breon (1995), which
combines the advantages of the DVI (NIR-red) and the NDVI for low and high vegetation cover, respectively:
$$RDVI = \frac{(\rho_{NIR} - \rho_{red})}{\sqrt{(\rho_{NIR} + \rho_{red})}} \qquad (4)$$
As a non-linear index, RDVI is not only less sensitive to variations in the geometrical and optical properties of
unknown foliage but also less affected by the solar and viewing geometry (Broge and Leblanc, 2001).
The NIR and SWIR bands are affected by the same ground properties, except that SWIR bands are also strongly
sensitive to equivalent water thickness. Fensholt and Sandholt (2003) proposed a vegetation index, the shortwave
infrared water stress index (SIWSI), using NIR and SWIR bands to estimate drought stress for vegetation in semi-arid
environments:
$$SIWSI_{12} = \frac{(\rho_{NIR} - \rho_{SWIR12})}{(\rho_{NIR} + \rho_{SWIR12})} \qquad (5)$$
$$SIWSI_{16} = \frac{(\rho_{NIR} - \rho_{SWIR16})}{(\rho_{NIR} + \rho_{SWIR16})} \qquad (6)$$
where $\rho_{swir12}$ is NBAR band 5 (1230-1250 nm) and $\rho_{swir16}$ is NBAR band 6 (1628-1652 nm). As the vegetation water
content increases, the reflectance in the SWIR decreases indicating that low and high SIWSI values point to sufficient
water conditions and drought stress, respectively.

**2.3.3 Incoming PAR across the Sahel**
Incoming PAR at the ground surface from ERA Interim followed the pattern of PAR measured at the six sites in situ
closely, but it was systematically underestimated (Fig. 2). An ordinary least square linear regression was thereby fitted
between ERA Interim PAR and PAR measured in situ ($PAR_{in\ situ}=3.09*\ PAR_{ERA\ interim} +23.07$; coefficient of
determination ($R^2$)=0.93; n=37976). The regression line was used for converting ERA Interim PAR to the same level as
in situ measured PAR.
<Figure 2>

**2.4 Data analysis**
**2.4.1 Coupling temporal and spatial dynamics in photosynthetic capacity and quantum efficiency with**
**explanatory variables**



In a first step, the coupling between intra-annual dynamics in $F_{opt}$ and $\alpha$ and the vegetation indices for the different
measurement sites were studied using Pearson correlation analysis. Relationships between intra-annual dynamics in $F_{opt}$
and $\alpha$ and the vegetation indices for all sites combined were also analysed. In order to avoid influence of the spatial and
inter-annual variability, time series of ratios of $F_{opt}$ and $\alpha$ ($F_{opt\_frac}$ and $\alpha_{frac}$) against the annual peak values ($F_{opt\_peak}$ and
$\alpha_{peak}$; see below for calculation of annual peak values) were estimated for all sites:
$$F_{opt\_frac} = \frac{F_{opt}}{F_{opt\_peak}} \tag{7}$$
$$\alpha_{frac} = \frac{\alpha}{\alpha_{peak}} \tag{8}$$
The same standardisation procedure was used for all vegetation indices ($VI_{frac}$):
$$VI_{frac} = \frac{VI}{VI_{peak}} \tag{9}$$
where $VI_{peak}$ is the annual peak values of the vegetation indices (14 days running mean with highest annual value). Such
a standardisation gives fractions of how $F_{opt}$, $\alpha$ and VI varies over the season in relationship to the annual peak value,
and it removes the spatial and inter-annual variation, and mainly intra-annual dynamics remains. The coupling between
$\alpha_{frac}$ and $F_{opt\_frac}$ and the different $VI_{frac}$ were examined using Pearson correlation analysis for all sites. The robustness of
the correlation coefficients was estimated by using a bootstrap simulation with 200 iterations in the correlation analysis
(Richter et al., 2012).
In order to investigate spatial and inter-annual variability in $F_{opt}$ and $\alpha$ for the measurement sites, gaps needed to be
filled. Regression trees were used to fill gaps in the daily estimates of $F_{opt}$ and $\alpha$. One hundred tree sizes were chosen
based on 100 cross validation runs, and these trees were then used for estimating the $F_{opt}$ and $\alpha$ (De'ath and Fabricius,
2000). We used SWC, VPD, $T_{air}$, PAR, and the vegetation index with strongest correlation with intra-annual dynamics
as explanatory variables in the analysis. In the analysis for all sites, the same standardisation procedure as done for $F_{opt}$,
$\alpha$, and the vegetation indices was done for the hydrological and meteorological variables. The 100 $F_{opt}$ and $\alpha$ output
subsets from the regression trees were averaged and used for filling the gaps in $F_{opt}$ and $\alpha$.
To investigate spatial and inter-annual variability in $F_{opt}$ and $\alpha$ across the measurement sites of the Sahel, annual peak
values of $F_{opt}$ and $\alpha$ ($F_{opt\_peak}$ and $\alpha_{peak}$; 14 days running mean with highest annual value) were correlated with the annual
sum of P, yearly means of $T_{air}$, SWC, RH, VPD, $R_g$, annual peak values of biomass, soil nitrogen and C concentrations,
C3/C4 ratio, and $VI_{peak}$ using Pearson linear correlations. Again, we used a bootstrap simulation methodology with 200
iterations in order to estimate the robustness of the correlations.

### 2.4.2 The GPP model

Based on Eq. 1 and the outcome of the statistical analysis previously described under subsection 2.4.1 (for results see
subsect. 3.2), a model for estimating GPP across the Sahel was created:
$$GPP = -F_{opt} \times \left(1 - e^{\left(\frac{-\alpha \times PAR}{F_{opt}}\right)}\right) \tag{10}$$



The model is applicable for each point in space and time. Firstly, $F_{opt\_peak}$ and $\alpha_{peak}$ were estimated spatially and inter-
annually using linear regression functions fitted against the vegetation indices with the strongest relationships to spatial
and inter-annual variability in $F_{opt\_peak}$ and $\alpha_{peak}$:
$$F_{opt\_peak} = k_{Fopt} \times NDVI_{peak} + m_{Fopt} \qquad (11)$$
$$\alpha_{peak} = k_\alpha \times RDVI_{peak} + m_\alpha \qquad (12)$$
where $k_{Fopt}$ and $k_\alpha$ are the slopes of the lines and $m_{Fopt}$ and $m_\alpha$ are the intercepts. Secondly, to estimate the $F_{opt\_frac}$ and
$\alpha_{frac}$ for each day of the year, linear regression functions were established for $F_{opt\_frac}$ and $\alpha_{frac}$ with the vegetation index
with the strongest relationships to intra-annual variability of $F_{opt\_frac}$ and $\alpha_{frac}$ for all sites, as follows:
$$F_{opt\_frac} = l_{Fopt} \times RDVI_{frac} + n_{Fopt} \qquad (13)$$
$$\alpha_{frac} = l_\alpha \times RDVI_{frac} + n_\alpha \qquad (14)$$
where $l_{Fopt}$ and $l_\alpha$ are the slopes of the lines and $n_{Fopt}$ and $n_\alpha$ are the intercepts. Eq. 11-14 provide the relationships to
estimate $F_{opt}$ and $\alpha$ for any day of the year and for any point in space across the Sahel:
$$F_{opt} = F_{opt\_peak} \times F_{opt\_frac} = \left(k_{Fopt} \times NDVI_{max} + m_{Fopt}\right)\left(l_{Fopt} \times RDVI_{frac} + n_{Fopt}\right) \qquad (15)$$
$$\alpha = \alpha_{peak} \times \alpha_{frac} = \left(k_\alpha \times RDVI_{max} + m_\alpha\right)\left(l_\alpha \times RDVI_{frac} + n_\alpha\right) \qquad (16)$$
Eq. 15 and 16 can be put into Eq. 10 and GPP is thereafter estimated as:
$$GPP = -\left(F_{opt\_peak} \times F_{opt\_frac}\right) \times \left(1 - e^{\left(\frac{-\left(\alpha_{peak} \times \alpha_{frac}\right) \times PAR}{F_{opt\_peak} \times F_{opt\_frac}}\right)}\right) \qquad (17)$$
generating a final model as:
$$GPP = -\left(\left(k_{Fopt} \times NDVI_{max} + m_{Fopt}\right)\left(l_{Fopt} \times RDVI_{frac} + n_{Fopt}\right)\right)$$
$$\times \left(1 - e^{\left(\frac{-\left(k_\alpha \times NDVI_{max} + m_\alpha\right)\left(l_\alpha \times RDVI_{frac} + n_\alpha\right) \times PAR}{\left(k_{Fopt} \times NDVI_{max} + m_{Fopt}\right)\left(l_{Fopt} \times RDVI_{frac} + n_{Fopt}\right)}\right)}\right) \qquad (18)$$


### 272 2.4.3 Parameterisation and evaluation of modelled GPP and evaluation of the MODIS GPP product

In order to estimate the robustness of the GPP model and its parameters, we used a bootstrap simulation methodology
when fitting the empirical relationships. We used 200 iterations and different measurement sites were used in the
different runs when fitting the empirical relationships (Richter et al., 2012). The runs generated 200 sets of slopes,
intercepts, and $R^2$, from which the medians and the standard deviations were estimated. Possible errors (e.g. random
sampling errors, aerosols, electrical sensor noise, filtering and gap-filling errors, clouds, and satellite sensor
degradation) can be present in both the predictor and the response variables. Hence, we selected reduced major axis
linear regressions to account for errors in both predictor and response variables when fitting the regression functions.
The regression models were validated against the left-out subsamples within the bootstrap simulation methodology by
calculating the root-mean-square-error (RMSE), and by fitting an ordinary least squares linear regression between
modelled and in situ variables.
Similarly, the MODIS GPP product (MOD17A2H, collection 6) was evaluated against in situ GPP by calculating
RMSE, and by fitting an ordinary least squares linear regression.



## 3 Results

### 3.1 Evaluation of the MODIS GPP product

There was a strong linear relationship between the MODIS GPP product (MOD17A2H; collection 6) and the in situ GPP (slope 0.17; intercept 0.11 g C m$^{-2}$ d$^{-1}$; R$^2$ 0.69; n=598). However, MOD17A2H strongly underestimated in situ GPP (Fig. 3) resulting in high RMSE (2.69 g C m$^{-2}$ d$^{-1}$).

<Figure 3>

### 3.2 Intra-annual dynamics in photosynthetic capacity and quantum efficiency

Intra-annual dynamics in $F_{opt}$ and α differed in amplitude, but were otherwise similar across the measurement sites in the Sahel (Fig. 4). There is no green ground vegetation during the dry season, and the low photosynthetic activity is due to few evergreen trees. This results in low values for both $F_{opt}$ and α during the dry season. The vegetation responded strongly to rainfall, and both $F_{opt}$ and α increased during the early phase of the rainy season. Generally, $F_{opt}$ peaked slightly earlier than α (average± 1 standard deviation: 7±10 days) (Fig. 4).

<Figure 4>

All vegetation indices described intra-annual dynamics in $F_{opt}$ well for all sites (Table 2). SIWSI$_{12}$ had the highest correlation for all sites except Wankama Millet, where it was RDVI. When all sites were combined, all indices described seasonality in $F_{opt}$ well, but RDVI had the strongest correlation (Table 2).

The intra-annual dynamics in α were also closely coupled to intra-annual dynamics in the vegetation indices for all sites (Table 2). For α, RDVI was the strongest index describing intra-annual dynamics, except for Wankama Fallow where it was EVI. When all sites were combined all indices described intra-annual dynamics in α well, but RDVI was still the index with the strongest relationship (Table 2).

<Table 2>

The regression trees used for gap-filling explained the intra-annual dynamics in $F_{opt}$ and α well for all sites (Table 3). The main explanatory variables coupled to intra-annual dynamics in $F_{opt}$ for all sites across the Sahel were in the order of RDVI, SWC, VPD, $T_{air}$, and PAR; and for α they were RDVI, SWC, VPD and $T_{air}$. For all sites across Sahel, incorporating hydrological and meteorological variables increased the ability to determine intra-annual dynamics in $F_{opt}$ and α compared to the ordinary least squares linear regressions against the RDVI (Table 2, data given as *r;* Table 3). The incorporation of these variables increased the R$^2$ from 0.81 to 0.87 and from 0.74 to 0.84, for $F_{opt}$ and α respectively.

<Table 3>

### 3.3 Spatial and inter-annual dynamics in photosynthetic capacity and quantum efficiency

Large spatial and inter-annual variability in $F_{opt\_peak}$ and $α_{peak}$ were found across the six measurement sites in the Sahel (Table 4). The average two week running mean peak values of $F_{opt}$ and α for all sites were 26.4 μmol CO$_2$ m$^{-2}$ s$^{-1}$ and 0.040 μmol CO$_2$ μmol PAR$^{-1}$, respectively. However, the ranges were large; $F_{opt\_peak}$ ranged between 10.1 μmol CO$_2$ m$^{-2}$ s$^{-1}$ (Wankama Millet 2005) and 50.0 μmol CO$_2$ m$^{-2}$ s$^{-1}$ (Dahra 2010), and $α_{peak}$ ranged between 0.020 μmol CO$_2$ μmol PAR$^{-1}$ (Demokeya 2007) and 0.064 μmol CO$_2$ μmol PAR$^{-1}$ (Dahra 2010) (Table 4). All vegetation indices determined spatial and inter-annual dynamics in $F_{opt\_peak}$ and $α_{peak}$ well (Table 5). NDVI$_{peak}$ was most closely coupled with $F_{opt\_peak}$ whereas RDVI$_{peak}$ was closest coupled with $α_{peak}$ (Fig. 5). $F_{opt\_peak}$ also correlated well with peak





dry weight biomass, C content in the soil, and RH, whereas $\alpha_{peak}$ also correlated well with peak dry weight biomass, and
C content in the soil (Table 5).
<Table 4>
<Table 5>
<Figure 5>

**3.4 Evaluation of the GPP model**
Modelled GPP was similar to in situ GPP on average, and there was a strong linear relationship between modelled GPP
and in situ GPP for all sites (Fig. 6; Table 6).  However, when separating the evaluation between measurement sites, it
can be seen that the model reproduced some sites better than others (Figure 7; Table 6). Wankama Millet is generally
overestimated whereas the model works on average well for Demokeya but underestimates high values (Fig. 7; Table
6). Variability of in situ GPP at the other sites is well reproduced by the model (Fig. 7; Table 6).  The final parameters
of the GPP model (Eq. 18) are given in Table 7.
<Figure 6>
<Figure 7>
< Table 6>
< Table 7>

**4 Discussion**
Vegetation productivity of semi-arid savanna ecosystems is primarily driven by intra-annual rainfall distribution
(Eamus et al., 2013; Brümmer et al., 2008; Moncrieff et al., 1997), and in the Sahel soil moisture conditions at the early
rainy season are especially important (Rockström and de Rouw, 1997; Tagesson et al., 2016; Mbow et al., 2013). We
thereby hypothesised that vegetation indices closely related to equivalent water thickness (SIWSI) would be strongly
linked to intra-annual dynamics in $F_{opt}$ and $\alpha$. Our hypothesis was not rejected for $F_{opt}$, since this was also the case for
all sites except for Wankama Millet (Table 2). The Wankama millet is a cropped agricultural site whereas all other sites
are savanna ecosystems. However, our hypothesis was rejected for $\alpha$, since it was more closely related to vegetation
indices related to chlorophyll abundance (RDVI and EVI). Leaf area index increases over the growing season and it is
closely related to the vegetation indices coupled with chlorophyll abundance (Tagesson et al., 2009). This increases the
canopy level quantum efficiency ($\alpha$) which explains the close relationship of $\alpha$ to RDVI. However, $F_{opt}$ peaked earlier in
the rainy season than $\alpha$ (Fig. 4).  Vegetation during this phase is vulnerable to drought conditions explaining the close
relationship of $F_{opt}$ to SIWSI. $F_{opt}$ can only increase up to a certain level due to other constraining factors (nutrient,
water and meteorological conditions) which could explain its closer relationship with $SIWSI_{12}$ than with RDVI.

We hypothesised that remotely sensed vegetation indices closely related to chlorophyll abundance can be used for

quantifying spatial and inter-annual dynamics in $F_{opt}$ and $\alpha$. Indeed, NDVI, EVI, and RDVI all had close correlations
with the spatial and inter-annual dynamics in $F_{opt}$ and $\alpha$ (Table 5). It was surprising that $NDVI_{peak}$ had the strongest
correlation with spatial and inter-annual variability for $F_{opt}$ (Table 5). Both EVI and RDVI should be less sensitive to
saturation effects than NDVI (Huete et al., 2002; Roujean and Breon, 1995), and based on this we assumed that peak
values of these indices should have stronger relationships to peak values of $F_{opt}$ and $\alpha$. However, vegetation indices with
a high sensitivity to changes in green biomass at high biomass loads, gets less sensitive to green biomass changes at low



biomass loads (Huete et al., 2002). Peak leaf area index for ecosystems across the Sahel is approximately 2, whereas the
saturation issue of NDVI generally starts at an leaf area index of about 2-5 (Haboudane et al., 2004). Additionally,
atmospheric scattering is much higher in the shorter wavelengths making EO-based vegetation indices including blue-
band information very sensitive to the atmospheric correction (Fensholt et al., 2006b), possibly explaining the lower
correlation for EVI.
Our model substantially overestimates GPP for Wankama Millet (Fig. 7f). As a crop field, this site differs in particular
from the other studied sites by its species composition, ecosystem structure, as well as land and vegetation management.
Crop fields in southwestern Niger are generally characterized by a rather low productivity resulting from decreased
fertility and soil loss caused by intensive land use (Cappelaere et al., 2009)**.** These specifics of the Wankama Millet site
may cause the model parameterised with observations from the other study sites to overestimate GPP at this site. The
model parameterised using observation from the other measurement sites without this strong anthropogenic influence
thus overestimates GPP. Similar results were found by Boulain et al. (2009) when applying an up-scaling model using
leaf area index for Wankama Millet and Wankama Fallow. It worked well for Wankama fallow whereas it was less
conclusive for Wankama Millet. The main explanation was low leaf area index in millet fields because of a low density
of millet stands due to agricultural practice. There is extensive savanna clearing for food production in the Sahel
(Leblanc et al., 2008; Boulain et al., 2009; Cappelaere et al., 2009). To further understand the impacts of this land cover
change on vegetation productivity and land atmosphere exchange processes, it is of urgent need for more study sites
covering cropped areas in this region.
In Demokeya, GPP is slightly underestimated for the year 2008 (Fig. 7c) because modelled $F_{opt}$ (the thick black line in
Fig. 5) is much lower than the actual measured value in 2008. An improvement of the model could be to incorporate
some parameters that constrain or enhance $F_{opt}$ depending on environmental stress. Indeed, the regression tree analysis
indicated that incorporating climatic and hydrological variables increased the ability to predict both $F_{opt}$ and α. On the
other hand, for spatial upscaling purposes, it has been shown that including modelled climatic constraints on LUE
decreases the ability to predict vegetation productivity due to the incorporated uncertainty in these modelled
meteorological variables (Fensholt et al., 2006a; Ma et al., 2014). For spatial upscaling to regional scales it is therefore
better to simply use relationships to EO data. This is particularly the case for the Sahel, one of the largest dryland areas
in the world that is characterised by few sites of meteorological observations.
Although MOD17A2 GPP has previously been shown to capture GPP relatively well in several different ecosystems
(Turner et al., 2006; Turner et al., 2005; Heinsch et al., 2006; Sims et al., 2006; Kanniah et al., 2009), it has been shown
to be underestimated for others (Coops et al., 2007; Gebremichael and Barros, 2006; Sjöström et al., 2013). GPP of
Sahelian drylands have not been well captured by MOD17A2 (Sjöström et al., 2013; Fensholt et al., 2006a), and as we
have shown, this underestimation persists in the latest MOD17A2H GPP (collection 6) product. The main reason for
this major underestimation is that maximum LUE is set to 0.84 g C MJ$^{-1}$ (open shrubland; Demokeya) and 0.86 g C MJ$^{-}$
$^{1}$ (grassland; Agofou, Dahra, Kelma; Wankama Millet and Wankama Fallow) in the BPLUT, i.e. much lower than
maximum LUE measured at the Sahelian measurement sites of this study (average: 2.47 g C MJ$^{-1}$; range: 1.58-3.50 g C
MJ$^{-1}$) (Sjöström et al., 2013; Tagesson et al., 2015a), a global estimate of ~1.5 g C MJ$^{-1}$ (Garbulsky et al., 2010), and a
savanna site in Australia (1.26 g C MJ$^{-1}$) (Kanniah et al., 2009).
Several state of the art dynamic global vegetation models have been used for decades to quantify GPP at different
spatial and temporal scales (Dickinson, 1983; Sellers et al., 1997). These models are generally based on the





photosynthesis model by Farquhar et al. (1980), a model particularly sensitive to uncertainty in photosynthetic capacity
(Zhang et al., 2014). This and several previous studies have shown that both photosynthetic capacity and efficiency
(both α and LUE) can considerably vary seasonally and spatially, both within and between vegetation types (Eamus et
al., 2013; Garbulsky et al., 2010; Ma et al., 2014; Tagesson et al., 2015a). This variability is difficult to estimate using
broad values based on land cover classes, yet most models apply a constant value which can cause substantial
inaccuracies in the estimates of seasonal and spatial variability in GPP. This is particularly a problem in savannas that
comprises of several plant functional types (C3 and C4 species, and a large variability in tree/herbaceous vegetation
fractions) (Scholes and Archer, 1997). This study indicates the strong applicability of EO as a tool for parameterising
spatially explicit estimates of plant physiological variables, which could improve our ability to simulate GPP. Spatially
explicit estimates of GPP at a high temporal and spatial resolution are essential for current global change studies and
would be advantageous in the analysis of changes in GPP, its relationship to climatic change and anthropogenic forcing,
and estimations of ecosystem processes and biochemical and hydrological cycles.

**Acknowledgements** Data is available from Fluxnet (http://fluxnet.ornl.gov) and CarboAfrica
(http://www.carboafrica.net/index en.asp). Data for the Mali and Niger sites were made available by the AMMA-
CATCH regional observatory (www.amma-catch.org), which is funded by the French Institut de Recherche pour le
Développement (IRD) and Institut National des Sciences de l'Univers (INSU). The project was funded by the Danish
Council for Independent Research (DFF) Sapere Aude programme. Faculty of Science, Lund University supported the
Dahra and Demokeya measurements with an infras-tructure grant. Ardö received support from the Swedish National
Space Board.

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



**Tables**
**Table 1.** Description of the six measurement sites including location, soil type, ecosystem type and dominant species.

| Measurement site | Coordinates | Soil type | Ecosystem | Dominant species |
|---|---|---|---|---|
| Agoufou[a] (ML-AgG, Mali) | 15.34°N, 1.48°W | Sandy ferruginous Arenosol | Open woody savanna (4% tree cover) | Trees: *Acacia spp., Balanites aegyptiaca, Combretum glutinosum* Herbs: *Zornia glochidiata, Cenchrus biflorus, Aristida mutabilis, Tragus berteronianus* |
| Dahra[b] (SN-Dah, Senegal) | 15.40°N, 15.43°W | Sandy luvic arenosol | Grassland/shrubland Savanna (3% tree cover) | Trees: *Acacia spp., Balanites aegyptiaca* Herbs: *Zornia latifolia, Aristida adscensionis, Cenchrus biflorus* |
| Demokeya[c] (SD-Dem, Sudan) | 13.28°N, 30.48°E | Cambic Arenosol | Sparse acacia savannah (7% tree cover) | *Trees: Acacia spp.,* Herbs: *Aristida pallida, Eragrostis tremula, Cenchrus biflorus* |
| Kelma[a] (ML-Kem, Mali) | 15.22°N, 1.57°W | Clay soil depression | Open acacia forest (90% tree cover) | Trees: *Acacia seyal, Acacia nilotica, Balanites aegyptiaca* Herbs: *Sporobolus hevolvus, Echinochloa colona, Aeschinomene sensitive* |
| Wankama Fallow[d] (NE-WaF, Niger) | 13.65°N, 2.63°E | Sandy ferruginous Arenosol | Fallow bush | *Guiera senegalensis* |
| Wankama Millet[e] (NE-WaM, Niger) | 13.64°N, 2.63°E | Sandy ferruginous Arenosol | Millet crop | *Pennisetum glaucum* |

[a](Timouk et al., 2009)
[b](Tagesson et al., 2015b)
[c](Sjöström et al., 2009)
[d](Velluet et al., 2014)
[e](Boulain et al., 2009)

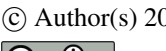



**Table 2.** Correlation between dynamics in photosynthetic capacity ($F_{opt}$), quantum efficiency ($\alpha$), and the different vegetation indices for the six measurement sites (Fig. 1). Values are averages±1 standard deviation from 200 bootstraping runs. The bold values are the indices with the strongest correlation. EVI is the enhanced vegetation index, NDVI is the normalized difference vegetation index, RDVI is the renormalized difference vegetation index, SIWSI is the shortwave infrared water stress index. $SIWSI_{12}$ is based on the MODIS Bidirectional Reflectance Distribution Functions (NBAR) band 2 and band 5, whereas $SIWSI_{16}$ is based on MODIS NBAR band 2 and band 6.

| Measurement site | $F_{opt}$ | | | | | $\alpha$ | | | | |
|---|---|---|---|---|---|---|---|---|---|---|
| | EVI | NDVI | RDVI | $SIWSI_{12}$ | $SIWSI_{16}$ | EVI | NDVI | RDVI | $SIWSI_{12}$ | $SIWSI_{16}$ |
| ML-AgG | 0.89±0.02 | 0.87±0.02 | 0.95±0.01 | **-0.95±0.01** | -0.93±0.02 | 0.92±0.02 | 0.91±0.01 | **0.96±0.01** | -0.94±0.01 | -0.88±0.02 |
| SN-Dah | 0.92±0.005 | 0.91±0.01 | 0.96±0.003 | **-0.96±0.004** | -0.93±0.01 | 0.89±0.01 | 0.90±0.01 | **0.93±0.01** | -0.92±0.01 | -0.87±0.01 |
| SD-Dem | 0.81±0.01 | 0.78±0.01 | 0.91±0.01 | **-0.93±0.01** | -0.90±0.01 | 0.76±0.02 | 0.73±0.02 | **0.86±0.01** | -0.82±0.02 | -0.79±0.02 |
| MA-Kem | 0.77±0.02 | 0.83±0.02 | 0.95±0.01 | **-0.95±0.01** | -0.90±0.02 | 0.69±0.05 | 0.73±0.04 | **0.80±0.03** | -0.77±0.03 | -0.76±0.03 |
| NE-WaF | 0.87±0.02 | 0.81±0.02 | 0.78±0.02 | **-0.90±0.01** | -0.80±0.02 | **0.89±0.01** | 0.84±0.01 | 0.85±0.01 | -0.88±0.01 | -0.79±0.01 |
| NE-WaM | 0.41±0.05 | 0.50±0.04 | **0.72±0.03** | -0.55±0.04 | -0.43±0.05 | 0.72±0.02 | 0.76±0.02 | **0.81±0.01** | -0.75±0.01 | -0.72±0.01 |
| All sites | 0.86±0.0 | 0.79±0.0 | **0.90±0.0** | 0.75±0.0 | 0.70±0.0 | 0.83±0.01 | 0.80±0.01 | **0.86±0.01** | 0.62±0.01 | 0.54±0.01 |



**Table 3.** Statistics for the regression tree analysis studying relationships between intra-annual dynamics in the the photosynthetic capacity ($F_{opt}$) and quantum efficiency (α) and the explanatory variables for the six measurement sites (Fig. 1). The pruning level is the number of splits of the regression tree and an indication of complexity of the system.

| Measurement site | Explanatory variables: | | | | | Pruning level | $R^2$ |
|---|---|---|---|---|---|---|---|
| $F_{opt}$ | 1 | 2 | 3 | 4 | 5 | | |
| ML-AgG | $SIWSI_{12}$ | Tair | PAR | SWC | | 16 | 0.98 |
| SN-Dah | $SIWSI_{12}$ | SWC | VPD | Tair | PAR | 84 | 0.98 |
| SD-Dem | $SIWSI_{12}$ | VPD | SWC | Tair | PAR | 33 | 0.97 |
| ML-Kem | $SIWSI_{12}$ | PAR | Tair | VPD | | 22 | 0.98 |
| NE-WaF | $SIWSI_{12}$ | SWC | VPD | Tair | | 14 | 0.92 |
| NE-WaM | RDVI | SWC | VPD | Tair | | 18 | 0.75 |
| All sites | RDVI | SWC | Tair | VPD | | 16 | 0.87 |
| α | | | | | | | |
| ML-AgG | RDVI | | | | | 3 | 0.95 |
| SN-Dah | RDVI | VPD | SWC | Tair | PAR | 21 | 0.93 |
| SD-Dem | RDVI | SWC | PAR | Tair | | 16 | 0.93 |
| ML-Kem | RDVI | Tair | | | | 4 | 0.75 |
| NE-WaF | EVI | SWC | VPD | | | 10 | 0.90 |
| NE-WaM | RDVI | SWC | VPD | Tair | | 15 | 0.86 |
| All sites | RDVI | SWC | VPD | Tair | | 16 | 0.84 |



**Table 4.** Annual peak values of quantum efficiency ($\alpha_{peak}$; µmol $CO_2$ µmol $PAR^{-1}$) and photosynthetic capacity ($F_{opt\_peak}$; µmol $CO_2$ $m^{-2}$ $s^{-1}$) for the six measurement sites (Fig. 1). The peak values are the 2 week running mean with highest annual value.

| Measurement site | Year | $\alpha_{peak}$ | $F_{opt\_peak}$ |
|---|---|---|---|
| ML-AgG | 2007 | 0.0396 | 24.5 |
| SN-Dah | 2010 | 0.0638 | 50.0 |
| | 2011 | 0.0507 | 42.3 |
| | 2012 | 0.0480 | 39.2 |
| | 2013 | 0.0549 | 40.0 |
| SD-Dem | 2007 | 0.0257 | 16.5 |
| | 2008 | 0.0327 | 21.0 |
| | 2009 | 0.0368 | 16.5 |
| ML-Kem | 2007 | 0.0526 | 33.5 |
| NE-WaF | 2005 | 0.0273 | 18.2 |
| | 2006 | 0.0413 | 21.0 |
| NE-WaM | 2005 | 0.0252 | 10.6 |
| | 2006 | 0.0200 | 10.1 |
| Average | | 0.0399 | 26.4 |



**Table 5.** Correlation matrix between annual peak values of photosynthetic capacity ($F_{opt\_peak}$) and quantum efficiency ($\alpha_{peak}$) and measured environmental variables. P is annual rainfall; $T_{air}$ is yearly averaged air temperature at 2 m height; SWC is yearly averaged soil water content (% volumetric water content) measured at 0.1 m depth; Rh is yearly averaged relative humidity; VPD is yearly averaged vapour pressure deficit; $R_g$ is yearly averaged incoming global radiation; N and C cont. are soil nitrogen and carbon contents; $NDVI_{peak}$ is annual peak normalized difference vegetation index (NDVI); $EVI_{peak}$ is annual peak enhanced vegetation index (EVI); $RDVI_{peak}$ is annual peak renormalized difference vegetation index (RDVI); $SIWSI_{12peak}$ is annual peak short wave infrared water stress index based on MODIS NBAR band 2 and band 5; and $SIWSI_{16peak}$ is annual peak short wave infrared water stress index based on MODIS NBAR band 2 and band 6. Sample size was 13 for all except the marked explanatory variables.

| Explanatory variable | $F_{opt\_peak}$ | $\alpha_{peak}$ |
|---|---|---|
| Meteorological data | | |
| P (mm) | $0.24\pm0.26$ | $0.13\pm0.27$ |
| $T_{air}$ (°C) | $-0.07\pm0.25$ | $-0.01\pm0.25$ |
| SWC (%)[a] | $0.33\pm0.25$ | $0.16\pm0.27$ |
| Rh (%) | $0.73\pm0.16^*$ | $0.60\pm0.19$ |
| VPD (hPa) | $0.20\pm0.26$ | $0.15\pm0.30$ |
| $R_g$ (W m$^{-2}$) | $-0.48\pm0.21$ | $-0.41\pm0.24$ |
| Biomass and edaphic data | | |
| Biomass (g DW m$^{-2}$)[a] | $0.77\pm0.15^*$ | $0.74\pm0.14^*$ |
| C3/C4 ratio | $-0.05\pm0.26$ | $0.06\pm0.30$ |
| N cont. (%)[b] | $0.22\pm0.11$ | $0.35\pm0.14$ |
| C cont. (%)[b] | $0.89\pm0.06^{**}$ | $0.87\pm0.07^{**}$ |
| Earth observation data | | |
| $NDVI_{peak}$ | $0.94\pm0.05^{**}$ | $0.87\pm0.07^{*}$ |
| $EVI_{peak}$ | $0.93\pm0.04^{**}$ | $0.87\pm0.07^{**}$ |
| $RDVI_{peak}$ | $0.93\pm0.04^{**}$ | $0.89\pm0.07^{**}$ |
| $SIWSI_{12peak}$ | $0.85\pm0.08^{**}$ | $0.84\pm0.08^{**}$ |
| $SIWSI_{16peak}$ | $0.67\pm0.12^*$ | $0.65\pm0.15^*$ |
| Photosynthetic variables | | |
| $F_{opt}$ | - | $0.94\pm0.03^{**}$ |

[a]sample size equals 11.
[b]sample size equals 9.
* significant at 0.05 level.
** significant at 0.01 level



**Table 6.** Statistics regarding the evaluation of the gross primary productivity (GPP) model for the six measurement sites (Fig. 1). In situ and modelled GPP are averages ± 1 standard deviation. RMSE is the root-mean-squares-error, and slope, intercept and $R^2$ is from the fitted ordinary least squares linear regression.

| Measurement site | In situ GPP ($\mu$mol $CO_2$ m$^{-2}$ s$^{-1}$) | Modelled GPP ($\mu$mol $CO_2$ m$^{-2}$ s$^{-1}$) | RMSE ($\mu$mol $CO_2$ m$^{-2}$ s$^{-1}$) | slope | Intercept ($\mu$mol $CO_2$ m$^{-2}$ s$^{-1}$) | $R^2$ |
|---|---|---|---|---|---|---|
| ML-AgG | 3.55±5.45 | 3.91±5.54 | 1.83±0.10 | 0.97±0.06 | 0.50±0.03 | 0.90±0.01 |
| SN-Dah | 9.14±10.12 | 8.60±10.72 | 3.85±1.34 | 0.99±0.07 | -0.44±1.11 | 0.87±0.04 |
| SD-Dem | 3.83±4.42 | 3.61±4.51 | 3.05±1.06 | 0.79±0.18 | 0.61±0.75 | 0.59±0.11 |
| ML-Kem | 11.17±7.98 | 10.73±10.50 | 5.06±1.23 | 1.16±0.20 | -2.29±1.65 | 0.79±0.12 |
| NE-WaF | 3.91±4.08 | 5.38±3.97 | 2.55±1.05 | 0.85±0.15 | 2.08±1.48 | 0.75±0.08 |
| NE-WaM | 2.25±2.00 | 5.51±3.93 | 4.13±0.99 | 1.63±0.45 | 1.84±1.01 | 0.68±0.05 |
| Average | 5.31±7.15 | 5.80±7.53 | 3.56±0.60 | 0.94±0.07 | 0.85±0.92 | 0.84±0.08 |





**Table 7.** The parameters for Eq. 18 that was used in the final gross primary productivity (GPP) model. RMSE is the root mean square error, and $R^2$ is the coefficient of determination of the linear regression models predicting the different variables.

| Parameter | Value | RMSE | $R^2$ |
|---|---|---|---|
| $k_{Fopt}$ | 79.6±6.3 | 5.1±1.3 | 0.89±0.05 |
| $m_{Fopt}$ | -7.3±3.2 | | |
| $l_{Fopt}$ | 1.81±0.07 | 0.33±0.04 | 0.79±0.04 |
| $n_{Fopt}$ | -0.85±0.07 | | |
| $k_{\alpha}$ | 0.16±0.02 | 0.0069±0.0021 | 0.81±0.10 |
| $m_{\alpha}$ | -0.014±0.007 | | |
| $l_{\alpha}$ | 1.20±0.05 | 0.38±0.04 | 0.71±0.04 |
| $n_{\alpha}$ | -0.98±0.06 | | |



**Figures**

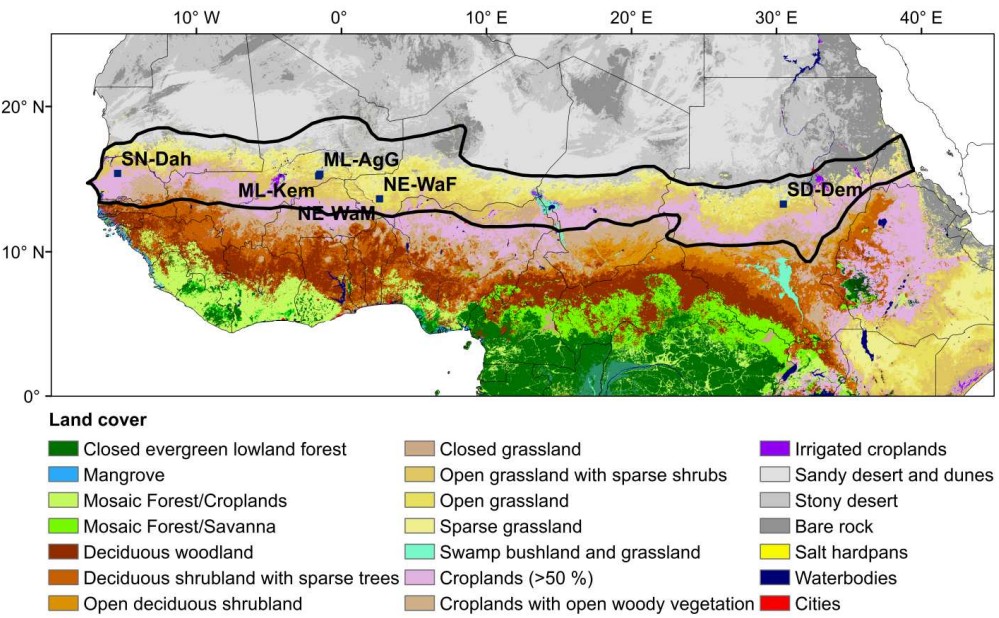

5  **Figure 1**. Land use cover classes for the Sahel and the location of the six measurement sites included in the study. The land cover classes are based on multi-sensor satellite observations (Mayaux et al., 2003). The sites are Agoufou (ML-AgG), Dahra (SN-Dah), Demokeya (SD-Dem), Kelma (ML-Kem), Wankama Fallow (NE-WaF), and Wankama Millet (NE-WaM). The thick black line is the borders of the Sahel based on the isohytes 150 and 700 mm of annual precipitation (Prince et al., 1995)





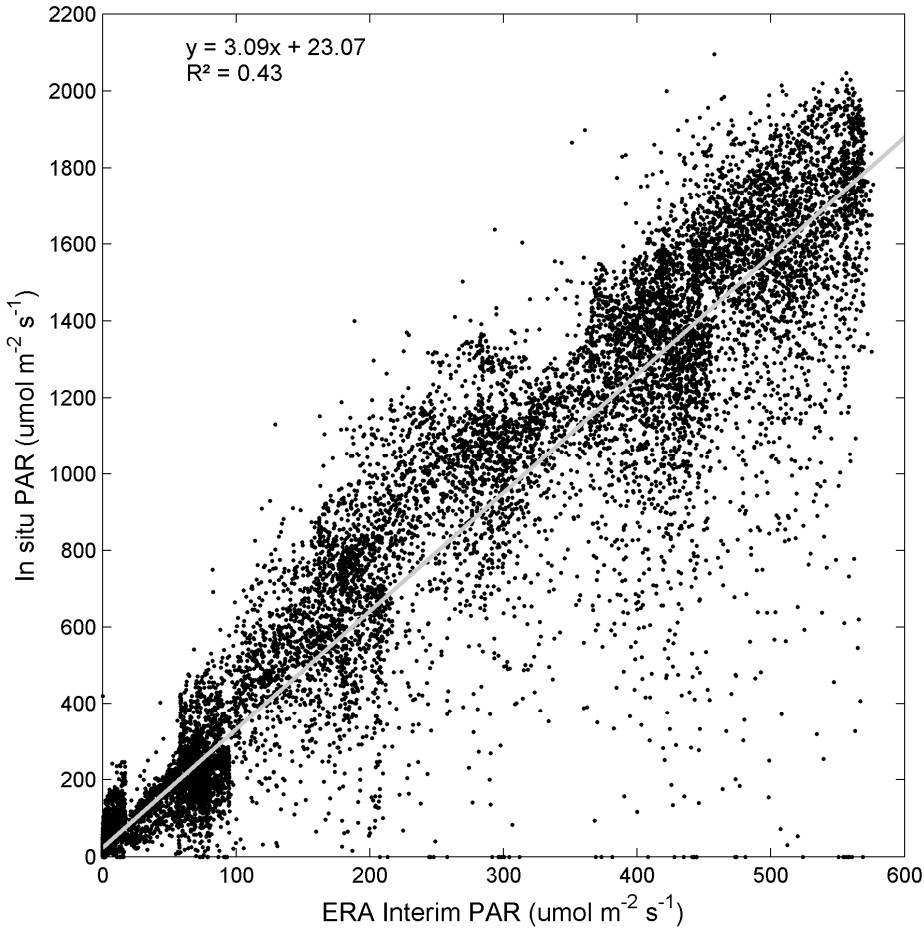

**Figure 2**. Photosynthetically active radiation (PAR) measured in situ against gridded ERA Interim ground surface PAR extracted for the six measurement sites (Figure 1) across the Sahel from European Centre for Medium-Range Weather Forecasts, ECMWF (2016). The grey line is the ordinary least square linear regression.





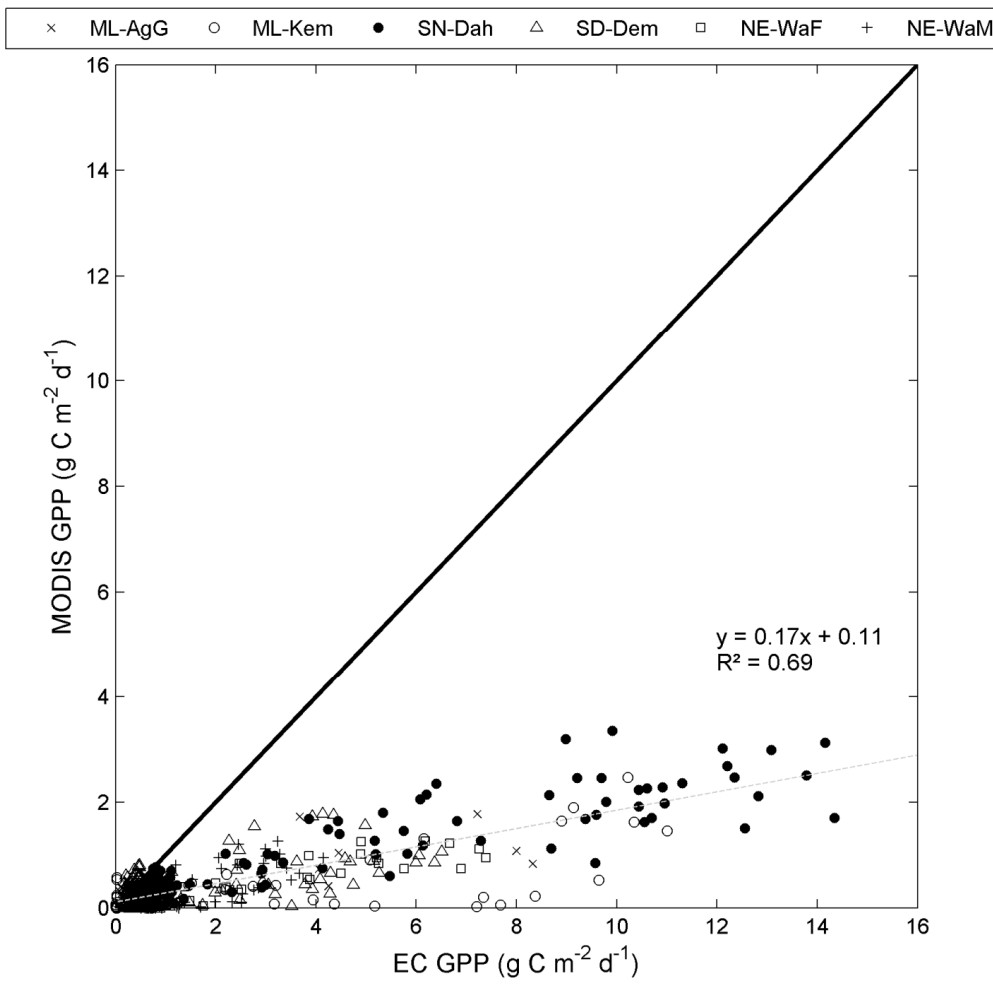

**Figure 3.** Evaluation of the MODIS based GPP product MOD17A2H collection 6 against eddy covariance based GPP from the six measurement sites (Figure 1) across the Sahel. The thick black line shows the one-to-one ratio, and the thin grey dotted line is the fitted ordinary least square linear regression.





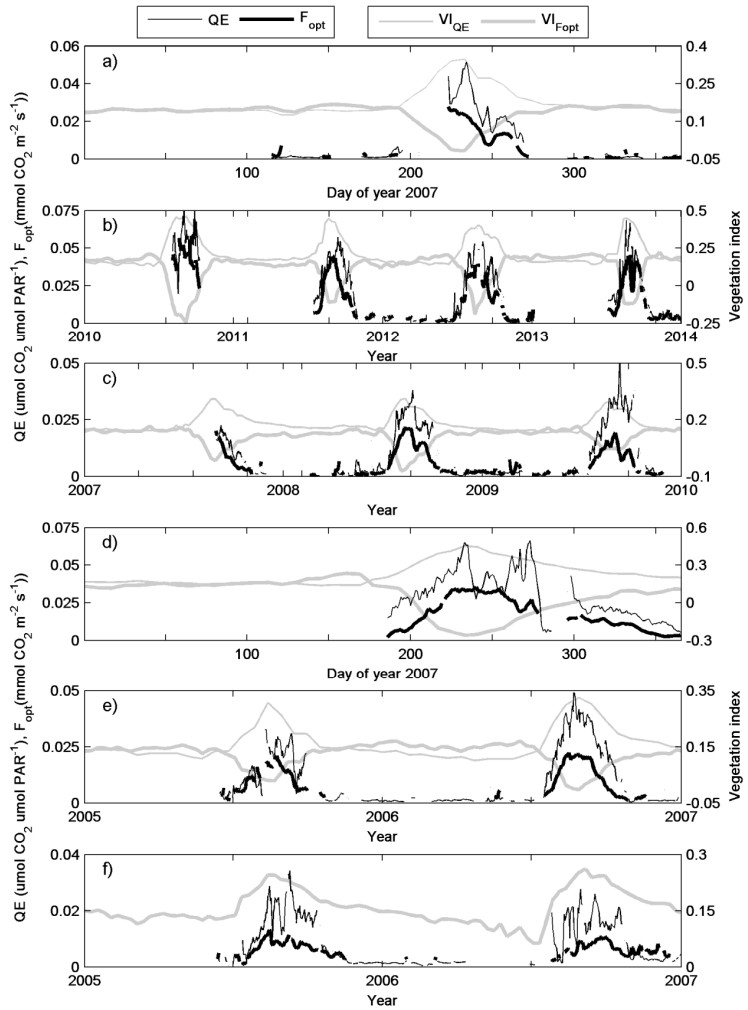

**Figure 4.** Dynamics in photosynthetic capacity ($F_{opt}$) and quantum efficiency (QE, α) for the six measurement sites. Included is also dynamics in the vegetation indices with highest correlation to the intra-annual dynamics in Fopt ($VI_{Fopt}$) and to quantum efficiency ($VI_{QE}$) (Table 2). The sites are a) Agoufou (ML-AgG), b) Dahra (SN-Dah), c) Demokeya (SD-Dem), d) Kelma (ML-Kem), e) Wankama Fallow (NE-WaF), and f) Wankama Millet (NE-WaM).





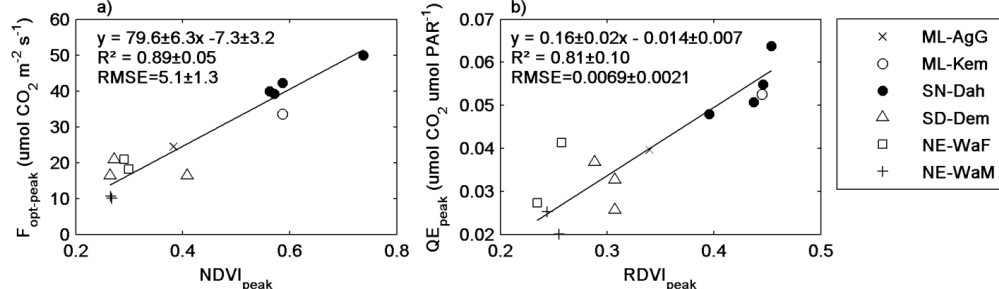

**Figure 5.** Scatter plots of annual peak values for the six measurement sites (Figure 1) of a) photosynthetic capacity ($F_{opt\_peak}$)

5    and b) quantum efficiency ($QE_{peak}$; $\alpha_{peak}$) against peak values of normalized difference vegetation index ($NDVI_{peak}$) and renormalized difference vegetation index ($RDVI_{peak}$), respectively. The annual peak values were estimated by taking the annual maximum of a two week running mean.



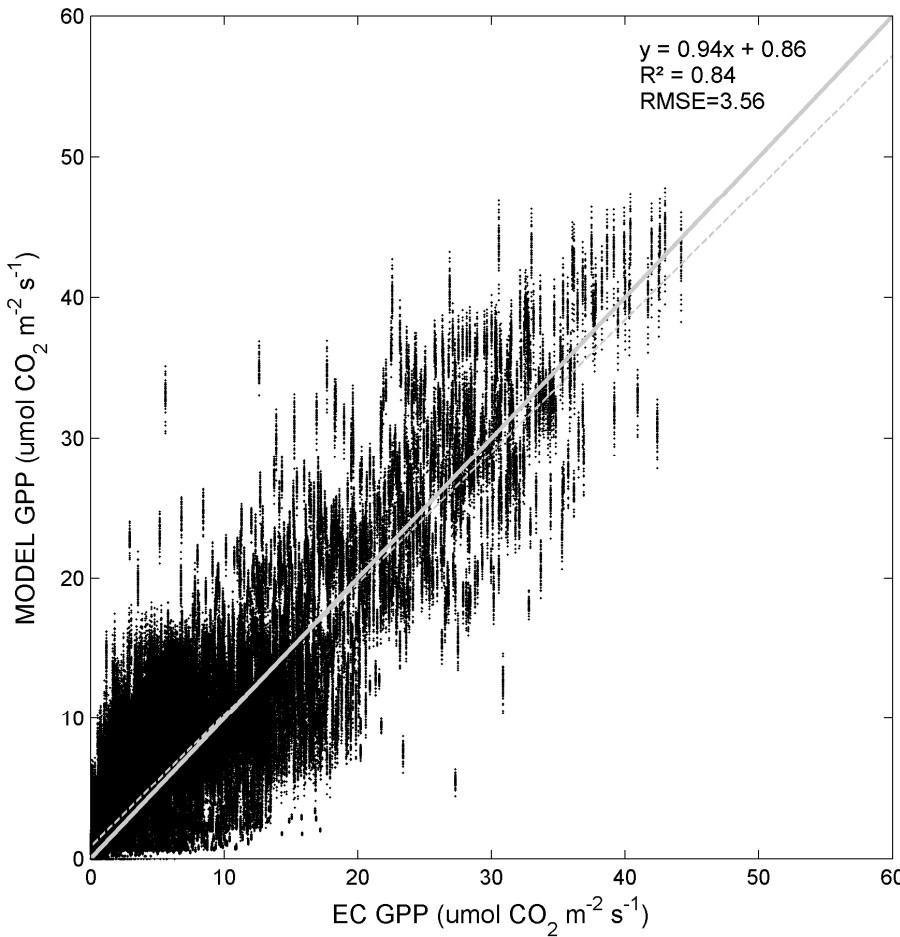

**Figure 6.** Evaluation of the modelled gross primary productivity (GPP) (Eq. 18) against in situ GPP from all six measurement sites across the Sahel. The thick grey line shows the one-to-one ratio, whereas the dotted thin grey line is the fitted ordinary least square linear regression.





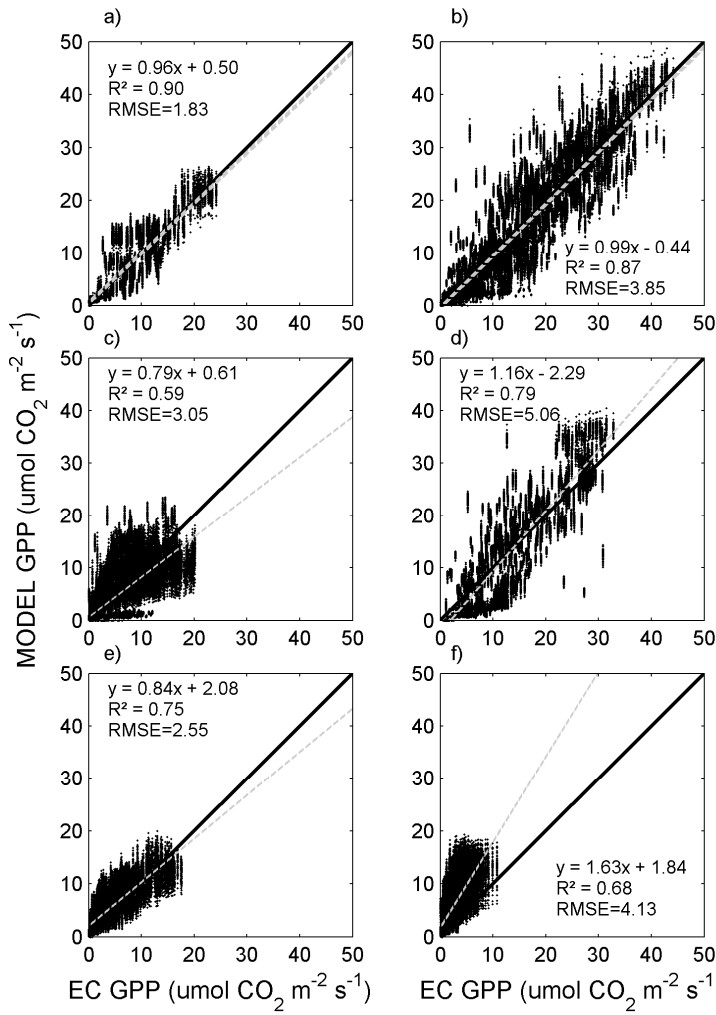

**Figure 7.** Evaluation of the modelled gross primary productivity (GPP) (Eq. 18) against in situ GPP for the six sites across Sahel (Figure 1). The thick black line shows the one-to-one ratio, whereas the dotted thin grey line is the fitted ordinary least square linear regression. The sites are a) Agoufou (ML-AgG), b) Dahra (SN-Dah), c) Demokeya (SD-Dem), d) Kelma (ML-Kem), e) Wankama Fallow (NE-WaF), and f) Wankama Millet (NE-WaM).