# Peer review of "Modelling spatial and temporal dynamics of GPP in the Sahel from"

_Biogeosciences, 2016_

## Referee Comment (RC1) · N. P. Hanan (Referee) · 24 Oct 2016

**A. Summary**

This paper uses data from six eddy covariance flux sites distributed across the Sahel of West Africa to examine patterns in space and time of carbon fluxes (GPP) as characterized by two key canopy-scale parameters (maximum photosynthetic uptake, called Fopt in this paper, and initial quantum yield, termed alpha). The authors also explore the relationships between the two GPP parameters and a variety of satellite vegetation indices providing (in theory at least) opportunities for spatial upscaling of the site-based results. This is an interesting paper reporting useful results.

[Figure]

B. Main Points

1. Regional GPP estimation. It is a pity the authors didn't take the final step to evaluate GPP across the region using the fitted models. At least, we don't see a map of these estimates, only point-based comparisons with the 6 field sites. In Section 2.4.1 the authors describe a "full model" for the regression tree used to characterize fluxes and predict Fopt and alpha at the field sites. In Section 2.4.2 they continue to describe an approach to derive parameters on a pixel-by-pixel basis where not all edaphic data (e.g. soil moisture) are available. However, we don't see the results of this analysis in the form of a map or other representation. Could this be added?

2. Prior work: The authors should refer to some considerable prior work that will be relevant to this analysis. See Global Change Biology 4, 523-538 (1998) and numerous HAPEX-Sahel papers in the J. Hydrology 1997 for earlier and quite detailed analysis of flux measurements in Sahelian vegetation. The GCB paper, for example, analyzes Fopt and alpha as a leaf-level variable in considerable detail. Note that the canopy-scale Fopt and alpha investigated here incorporate the effects of changing LAI during the season. This rather complicates the situation for this analysis, as the authors state on line 351.

3. Peak uptake rates: the field measurements at some sites seem abnormally high. The earlier data in the GCB paper references above was for a southern Sahel site with LAI likely higher than any of these sites, but with maximum Fopt of only ~15-20 umol m-2 s-1.

4. Possible unit issues: this is an impertinent question, but looking at the massive multipliers between the author's estimates and independent estimates in Figures 2 (incoming PAR) and 3 (GPP) I couldn't help wondering if there might be some unit issues. In the case of PAR the conversion of PAR in W/m2 to umol m-2 s-1 varies somewhat based on solar angle and atmospheric conditions but is typically 4.2 umol/W. This is more than the 3.09 of the fitted slope, but is it really possible that the ERA

PAR product is underestimating actual incoming PAR so consistently by a whopping 70% ! Similarly for Figure 3, if the MODIS product is in units of g/m2/day carbon and the authors have retained their data in units g/m2/day CO2 this would give an inherent slope in Figure 3 of 12/44 = 0.273. Again this doesn't entirely account for their calculated slope of 0.17, but might be worth double-checking.

C. Minor Points

Line 42: While it is appropriate to mention that significant inter-annual variability in global carbon cycle arises in semi-arid regions relating to rainfall variability and fire (particularly in the mesic savannas, more so than the Sahel; eg. Williams et al Carbon Balance and Management 2007), it would be an exaggeration to state that the semi-aris regions are "driving long-term trends".

Line 52: "continuous cropping" is very rare in the Sahel (outside of areas with irrigation opportunities, anyway). In the drier northern regions pastoralist communities may attempt a dryland crop, but with little expectation of success. Even in the wetter southern Sahel where the crop site in this paper is located, most fields are fallowed. In the highly populated regions near the capital city of Niger, rotations have reduced, but it would be wrong to imply that "continuous cropping is practiced" widely.

Line 107: "find evidence" is awkward here. Perhaps substitute "characterize".

––––––––––––––––––––––––––

---

## Referee Comment (RC2) · Anonymous Referee #2 · 2 Nov 2016

General comments: This is an interesting paper providing detailed descriptions of spatial and temporal dynamics in canopy light-response parameters at $CO_2$ flux observation sites across Sahel region. The authors evaluated MODIS GPP, and reported its serious problem. This paper demonstrated the applicability of alternative model to scale up EC flux-based GPP to regional or continental scales, using EO-based spectral vegetation indices. The dynamics of photosynthetic parameters and some interpretations of several vegetation indices presented in this paper are valuable to estimate $CO_2$ budget in semi-arid ecosystems, which have included large uncertainties so far. Overall presentation is well structured and clear. The purpose of this paper fits well to this journal.

Specific comments:

1. The intra-annual dynamics in $F_{opt}$ and $\alpha$ were well explained with the vegetation indices in relation to the seasonal changes in water thickness and chlorophyll abundance. But the shorter term variations in $F_{opt}$ and $\alpha$ (Fig. 4) do not seem to be explained sufficiently by the regression tree analysis. Some stress events may affect them. Please show the relationships with meteorological variables such as SWC or VPD additionally, and describe more information on the related specific stress events.

2. The result of strong underestimation of ERA Interim PAR against in situ PAR is surprising and important information. Please confirm the ERA Interim PAR data: it is W m$^{-2}$ (Line 157), but $\mu$mol m$^{-2}$ s$^{-1}$ (Fig.2). In addition, there seems to be some different tendencies in the relationships in Fig. 2, maybe depending on the periods and sites. Were the PAR sensors calibrated regularly? PAR sensors tend to deteriorate as aging. Please check the deterioration in PAR by comparison with the simultaneously measured Rg.

3. This paper aims to provide a model to scale up observed canopy scale GPP to regional or continental scales, using EO-based spectral vegetation indices. The readers will expect a final map of spatial distribution of GPP in semi-arid areas, and the map would make this paper more valuable.

Minor comments:

Line 184: What do you mean by "air-water interface"?

Table 2: Correlation between "intra-annual" dynamics

Please unify the descriptions: use $F_{opt_{frac}}$ and $\alpha_{frac}$ for intra-annual dynamics instead $F_{opt}$ and $\alpha$ in Table 2, 3, as described in the text.

Fig 3: Some points of ML-Kem are quite low (nearly 0) for MODIS GPP, while around 8 $\mu$mol m$^{-2}$ s$^{-1}$ for EC GPP. Why?

Please unify the descriptions: $\alpha$ instead of QE, as described in the text. Clarify the

labels and scales on X-axes.

(f) What is the reason that VI decreased less than 0.15 before the growing season in 2007 at NE-WaM?

---

## Editor Comment (EC1) · C. Bourque (Editor) · 4 Nov 2016

Thank you for your input. The authors should consider all of your comments in their revision of their manuscript.

Kind regards,

Charles Bourque

---

## Editor Comment (EC2) · C. Bourque (Editor) · 4 Nov 2016

I appreciate receiving your comments. The authors should strive to address all of your comments in their revision of the manuscript.

Kind regards,

Charles Bourque
* * *

---

## Author Comment (AC1) · 20 Dec 2016

Response to comments by Reviewer #1

A. Summary This paper uses data from six eddy covariance flux sites distributed across the Sahel of West Africa to examine patterns in space and time of carbon fluxes (GPP) as characterized by two key canopy-scale parameters (maximum photosynthetic uptake, called Fopt in this paper, and initial quantum yield, termed alpha). The authors also explore the relationships between the two GPP parameters and a variety of satellite vegetation indices providing (in theory at least) opportunities for spatial upscaling of the site-based results. This is an interesting paper reporting useful results.

[Figure]

Response: Thank you very much, and also thank you for insightful comments that helped improving the manuscript.

B. Main Points 1. Regional GPP estimation. It is a pity the authors didn't take the final step to evaluate GPP across the region using the fitted models. At least, we don't see a map of these estimates, only point-based comparisons with the 6 field sites. In Section 2.4.1 the authors describe a "full model" for the regression tree used to characterize fluxes and predict Fopt and alpha at the field sites. In Section 2.4.2 they continue to describe an approach to derive parameters on a pixel-by-pixel basis where not all edaphic data (e.g. soil moisture) are available. However, we don't see the results of this analysis in the form of a map or other representation. Could this be added?

Response: We agree with the reviewer; in the previous version of the manuscript we did not include the full gridded map because the spatial up-scaling requires some very heavy computer processing. However, we have now borrowed computer power from the university, and in the revised version of the manuscript we have included a full gridded map of peak Fopt, peak $\alpha$ and an annual sum of GPP.

2. Prior work: The authors should refer to some considerable prior work that will be relevant to this analysis. See Global Change Biology 4, 523-538 (1998) and numerous HAPEX-Sahel papers in the J. Hydrology 1997 for earlier and quite detailed analysis of flux measurements in Sahelian vegetation. The GCB paper, for example, analyses Fopt and alpha as a leaf-level variable in considerable detail. Note that the canopy scale Fopt and alpha investigated here incorporate the effects of changing LAI during the season. This rather complicates the situation for this analysis, as the authors state on line 351.

Response: Thank you for this suggestion, we agree that it was a good idea to extend the comparison of the results of our analysis to the results of previously published research. This has been incorporated into the revised manuscript.

3. Peak uptake rates: the field measurements at some sites seem abnormally high.

The earlier data in the GCB paper references above was for a southern Sahel site with LAI likely higher than any of these sites, but with maximum Fopt of only -15-20 umol m-2 s-1.

Response: 1) The leaf area index value of the HAPEX–Sahel West-Central fallow savanna site in (Hanan et al., 1998) is not larger than at the Dahra and Kelma sites, which are the two sites of our study with very high Fopt and $\alpha$. Peak LAI is 2.1 for Dahra and 2.7 for Kelma, so it is considerably higher than 1.2 as given in (Hanan et al., 1998). The higher LAI can thereby explain parts of the higher Fopt estimates. 2) (Hiernaux et al., 2009) and (Dardel et al., 2014) showed above ground peak biomass in southwestern Niger which are comparable, and nowadays slightly lower than what is reported for the Gourma area (which in addition receives less rain). Hanan et al 1997 (J Hydrology) report above ground peak biomass of 1000 and 1500 kg/ha for the grass and shrub fallow sites, which is much lower than what is reported for the Dahra site (Mbow et al., 2013), which also receives less rainfall. This is in line with a productivity gradient over these 3 sites, possibly caused by soil fertility and fallow management in southwestern Niger. 3) The reason for the high estimates of Fopt and $\alpha$ are the very high net CO2 fluxes measured by the eddy covariance systems. For the Dahra field site, we have performed a rigorous quality check of the data, please see (Tagesson et al., 2016) and we are certain that the measured values are correctly measured. Tagesson et al. (2016) have tried to explain the high net CO2 flux values by that there is a combination of dense herbaceous C4 ground vegetation, high soil nutrient availability, a grazing pressure resulting in compensatory growth and fertilization effects, and the West African Monsoon bring a humid layer of surface air from the Atlantic, possibly increasing vegetation productivity for the most western part of Sahel. This info has been included in the revised manuscript (L390-403).

4. Possible unit issues: this is an impertinent question, but looking at the massive multipliers between the author's estimates and independent estimates in Figures 2 (incoming PAR) and 3 (GPP) I couldn't help wondering if there might be some unit

issues. In the case of PAR the conversion of PAR in W/m2 to umol m-2 s-1 varies somewhat based on solar angle and atmospheric conditions but is typically 4.2 umol/W. This is more than the 3.09 of the fitted slope, but is it really possible that the ERA PAR product is underestimating actual incoming PAR so consistently by a whopping 70% ! Similarly for Figure 3, if the MODIS product is in units of g/m2/day carbon and the authors have retained their data in units g/m2/day CO2 this would give an inherent slope in Figure 3 of 12/44 = 0.273. Again this doesn't entirely account for their calculated slope of 0.17, but might be worth double-checking.

Response: Yes, we absolutely understand your concern here, and we have been looking at these conversions many times to make absolutely sure that the conversions are correctly done:

1. PAR values: The average raw in-situ PAR = 483 $\mu$mol m-2 s-1

The average raw ECMWF PAR = 350503 (J m-2 summed for 3 hours)

To get ECMWF PAR to (W m-2): raw ECMWF PAR was divided by (60sec*60 minutes*3 hours) => Average ECMWF PAR (W m-2) =350503/(60*60*3)= 32 W m-2.

To convert ECMWF PAR (W m-2) to $\mu$mol m-2 we multiplied with 4.57 (Sager and McFarlane, 1997):

Average ECMWF PAR ($\mu$mol m-2 s-1) =32*4.57= 148 $\mu$mol m-2 s-1

Average in-situ PAR ($\mu$mol m-2 s-1)/ Average ECMWF PAR ($\mu$mol m-2 s-1) = 483/148 = 3.2

So we think that the PAR conversion is correctly done. We recently found out that the issue is related to a major bug in the code of ECMWF surface PAR: "The surface incident value (code 58) seems erroneously low. For example, in locations in the Celtic Sea, surface PAR is typically around 20% to 25% of the clear sky value (code 20), and about a third of in-situ measurement of surface PAR. Cause: We have shortwave bands that include 0.442-0.625 micron, 0.625-0.778 micron and 0.778-1.24 micron. PAR is

coded as if it was intending to sum all of the radiation in the first of these and 0.42 of the second (to account for the fact that PAR is normally defined to stop at 0.7 microns. However, PAR is in fact calculated from the sum of the second band plus 0.42 of the third." (ECMWF, 2016). This indicates that the ERA-interim surface PAR product is actually not PAR, but rather incoming red and near infrared. However, we still intend to use this data source since we relate the gridded ECMWF PAR to in-situ measured PAR and used this relationship to convert ECMWF PAR to the proper level. The relationship should be ok, even if it is relating in-situ PAR to a different part of the spectrum; the final product is still PAR at a reasonable level. The issue is now described in the revised manuscript.

2. MODIS GPP: An example for GPP of Agofou:

Average in-situ GPP -1.34 $\mu$mol CO2 m-2 sec-1

Convert it to g CO2 m-2 and s-1:

1 mol=44 g CO2 and micro=$\mu$=10-6

ïČř Average in-situ GPP =0.000059 g CO2 m-2 s-1

Convert it to g CO2 m-2 and 8 d-1:

8 days = (8*24*60*60) seconds

0.000059 g CO2 m-2 s-1 * (8*24*60*60)

ïČř Average in-situ GPP =40.7 g CO2 m-2 and 8 day-1:

Convert it to g C m-2 and 8 d-1:

1 g CO2 = 0.27 g C

Average in-situ GPP = 40.7*0.27= 11.0 g C m-2 and 8 day-1:

Average raw MODIS GPP for Agofou: 24.1

Scaling factor: 0.0001 =>

Modis GPP (kg C m-2 and 8 day-1)=0.00241 kg C m-2 and 8 day-1

Modis GPP (g C m-2 and 8 day-1)=0.00241 *1000 = 2.41 g C m-2 and 8 day-1.

Again, we agree that this major underestimation is strange, but we believe that all conversions are correctly done.

C. Minor Points Line 42: While it is appropriate to mention that significant inter-annual variability in global carbon cycle arises in semi-arid regions relating to rainfall variability and fire (particularly in the mesic savannas, more so than the Sahel; eg. Williams et al Carbon Balance and Management 2007), it would be an exaggeration to state that the semiarid regions are "driving long-term trends".

Response: We agree with the reviewer that this was not a very clear sentence. But still, according (Ahlström et al., 2015) semi-arid region are driving the long term trends. We have clarified this in the revised manuscript:

"Vegetation growth in semi-arid regions is an important sink for human induced fossil fuel emissions. The mean carbon dioxide ($CO_2$) uptake by terrestrial ecosystems is dominated by highly productive lands, mainly tropical forests; wheras semi-arid regions are the main biome driving its inter-annual variability (Ahlström et al., 2015; Poulter et al., 2014). Semi arid regions also contribute to 60% of the long term trend in the global terrestrial C sink (Ahlström et al., 2015; Poulter et al., 2014)."

Line 52: "continuous cropping" is very rare in the Sahel (outside of areas with irrigation opportunities, anyway). In the drier northern regions pastoralist communities may attempt a dryland crop, but with little expectation of success. Even in the wetter southern Sahel where the crop site in this paper is located, most fields are fallowed. In the highly populated regions near the capital city of Niger, rotations have reduced, but it would be wrong to imply that "continuous cropping is practiced" widely.

Response: Thank you for noticing this, this sentence has been removed.

Line 107: "find evidence" is awkward here. Perhaps substitute "characterize".

Response: Yes, we fully agree. Characterize is much better. Thank you very much.

References Ahlström, A., Raupach, M. R., Schurgers, G., Smith, B., Arneth, A., Jung, M., Reichstein, M., Canadell, J. G., Friedlingstein, P., Jain, A. K., Kato, E., Poulter, B., Sitch, S., Stocker, B. D., Viovy, N., Wang, Y. P., Wiltshire, A., Zaehle, S., and Zeng, N.: The dominant role of semi-arid ecosystems in the trend and variability of the land CO2 sink, Science, 348, 895-899, 10.1126/science.aaa1668, 2015. Dardel, C., Kergoat, L., Hiernaux, P., Mougin, E., Grippa, M., and Tucker, C. J.: Re-greening Sahel: 30 years of remote sensing data and field observations (Mali, Niger), Remote Sens. Environ., 140, 350-364, http://dx.doi.org/10.1016/j.rse.2013.09.011, 2014. ECMWF: ERA-Interim: surface photosynthetically active radiation (surface PAR) values are too low https://software.ecmwf.int/wiki/display/CKB/ERA-Interim%3A+surface+photosynthetically+active+radiation+%28surface+PAR%29+values+are+too+low, access: 7 November, 2016. Hanan, N., Kabat, P., Dolman, J., and Elbers, J. A. N.: Photosynthesis and carbon balance of a Sahelian fallow savanna, Global Change Biol., 4, 523-538, 1998. Hiernaux, P., Mougin, E., Diarra, L., Soumaguel, N., Lavenu, F., Tracol, Y., and Diawara, M.: Sahelian rangeland response to changes in rainfall over two decades in the Gourma region, Mali, J. Hydrol., 375, 114-127, http://dx.doi.org/10.1016/j.jhydrol.2008.11.005, 2009. Mbow, C., Fensholt, R., Rasmussen, K., and Diop, D.: Can vegetation productivity be derived from greenness in a semi-arid environment? Evidence from ground-based measurements, J. Arid Environ., 97, 56-65, http://dx.doi.org/10.1016/j.jaridenv.2013.05.011, 2013. Poulter, B., Frank, D., Ciais, P., Myneni, R. B., Andela, N., Bi, J., Broquet, G., Canadell, J. G., Chevallier, F., Liu, Y. Y., Running, S. W., Sitch, S., and van der Werf, G. R.: Contribution of semi-arid ecosystems to interannual variability of the global carbon cycle, Nature, 509, 600-603, 10.1038/nature13376, 2014. Sager, J. C., and McFarlane, J. C.: Chapter 1. Radiation, in: Plant growth chamber handbook, edited by: W., L. R., and T.W., T., Iowa State University, Ames, 1997. Tagesson, T., Fensholt, R., Guiro, I., Cropley,

F., Horion, S., Ehammer, A., and Ardö, J.: Very high carbon exchange fluxes for a grazed semi-arid savanna ecosystem in West Africa, Danish Journal of Geography, 116, 93-109, http://dx.doi.org/10.1080/00167223.2016.1178072 2016.

Please also note the supplement to this comment:
http://www.biogeosciences-discuss.net/bg-2016-414/bg-2016-414-AC1-supplement.pdf

---

## Author Comment (AC2) · 20 Dec 2016

Response to comments by Reviewer #2

General comments: This is an interesting paper providing detailed descriptions of spatial and temporal dynamics in canopy light-response parameters at CO2 flux observation sites across Sahel region. The authors evaluated MODIS GPP, and reported its serious problem. This paper demonstrated the applicability of alternative model to scale up EC flux-based GPP to regional or continental scales, using EO-based spectral vegetation indices. The dynamics of photosynthetic parameters and some interpretations of several vegetation indices presented in this paper are valuable to estimate CO2 budget in semi-arid ecosystems, which have included large uncertainties so far. Overall presentation is well structured and clear. The purpose of this paper fits well to this journal.

Response: Thank you very much, and also thank you for insightful comments that helped improving our manuscript.

Specific comments: 1. The intra-annual dynamics in Fopt and $\alpha$ were well explained with the vegetation indices in relation to the seasonal changes in water thickness and chlorophyll abundance. But the shorter term variations in Fopt and $\alpha$ (Fig. 4) do not seem to be explained sufficiently by the regression tree analysis. Some stress events may affect them. Please show the relationships with meteorological variables such as SWC or VPD additionally, and describe more information on the related specific stress events.

Response: We are truly sorry, but we do not completely agree. In Table 3, results from the regression trees are presented and the coefficient of determination (R2) is larger than 0.9 for most sites; when all sites are combined it was 0.87 and 0.84 for Fopt and $\alpha$ respectively. So we would say that the regression trees describe the short term variability in Fopt and $\alpha$ pretty well. To further clarify this, we have incorporated a figure to the supplementary material with both measured and regression tree predicted Fopt and $\alpha$. This indicates that SWC and VPD have a strong influence on the short term variability, since these explanatory variables are included in most regression trees (Table 3). This info is included in the revised manuscript.

2. The result of strong underestimation of ERA Interim PAR against in situ PAR is surprising and important information. Please confirm the ERA Interim PAR data: it is W m-2 (Line 157), but $\mu$mol m-2 s-1 (Fig.2). In addition, there seems to be some different tendencies in the relationships in Fig. 2, maybe depending on the periods and sites. Were the PAR sensors calibrated regularly? PAR sensors tend to deteriorate as aging. Please check the deterioration in PAR by comparison with the simultaneously measured Rg.

[Figure]

Response: We completely understand your concern regarding this relationship, and we were very concerned ourselves. 1) Regarding the in-situ PAR data; we agree, two PAR sensors standing next to each other can easily give quite different values, and some minor differences between in-situ PAR and ECMWF could possibly be explained by this issue. However, the sensors have been sent for calibration regularly, and they have been intercalibrated before and after each rainy season. So this should not be a major issue. The different tendencies seen is most likely related to the fact that ECMWF PAR is given in UTC time for each 3h. We converted this to local time when comparing against the in-situ data, and different periods of the day thereby might get slightly different tendencies in the relationship. 2) Regarding the unit conversions: we have been looking at these conversions many times to make absolutely sure that the conversions are correctly done:

The average raw in-situ PAR = 483 $\mu$mol m-2 s-1

The average raw ECMWF PAR = 350503 (J m-2 summed for 3 hours)

To get ECMWF PAR to (W m-2): raw ECMWF PAR was divided by (60sec*60 minutes*3 hours) => Average ECMWF PAR (W m-2) =350503/(60*60*3)= 32 W m-2.

To convert ECMWF PAR (W m-2) to $\mu$mol m-2 we multiplied with 4.57 (Sager and McFarlane, 1997):

Average ECMWF PAR ($\mu$mol m-2 s-1) =32*4.57= 148 $\mu$mol m-2 s-1

Average in-situ PAR ($\mu$mol m-2 s-1)/ Average ECMWF PAR ($\mu$mol m-2 s-1) = 483/148 = 3.2

So we think that the PAR conversion is correctly done. We recently found out that the issue is related to a major bug in the code of ECMWF: "The surface incident value (code 58) seems erroneously low. For example, in locations in the Celtic Sea, surface PAR is typically around 20% to 25% of the clear sky value (code 20), and about a third of in-situ measurement of surface PAR. Cause: We have shortwave bands that include

0.442-0.625 micron, 0.625-0.778 micron and 0.778-1.24 micron. PAR is coded as if it was intending to sum all of the radiation in the first of these and 0.42 of the second (to account for the fact that PAR is normally defined to stop at 0.7 microns. However, PAR is in fact calculated from the sum of the second band plus 0.42 of the third." (ECMWF, 2016).

This indicates that the ERA-interim surface PAR product is actually not PAR, but rather incoming red and near infrared. However, we still intend to use this data source since we relate the gridded ECMWF PAR to in-situ measured PAR and used this relationship to convert ECMWF PAR to the proper level. The relationship should be ok, even if it is relating in-situ PAR to a different part of the spectrum; the final product is still PAR at a reasonable level. The issue is now described in the revised manuscript.

3. This paper aims to provide a model to scale up observed canopy scale GPP to regional or continental scales, using EO-based spectral vegetation indices. The readers will expect a final map of spatial distribution of GPP in semi-arid areas, and the map would make this paper more valuable.

Response: We agree with the reviewer, in the previous version we did not include the full gridded map because the spatial up-scaling requires some very heavy computer processing. However, we have now borrowed computer power from the university, and in the revised version of the manuscript we have included a full gridded map of peak Fopt, peak alpha and an annual sum of GPP.

Minor comments: Line 184: What do you mean by "air-water interface"?

Response: We agree that the formulation was not clear. This has been corrected in the revised manuscript:

"The NIR radiance is reflected by the leaf cells since an absorption of these wavelengths would result in overheating of the plant whereas red radiance is absorbed by chlorophyll and its accessory pigments (Gates et al., 1965)."

Table 2: Correlation between "intra-annual" dynamics

Response: Thank you for pointing this out. This has been taken care of.

Please unify the descriptions: use Foptf rac and _frac for intra-annual dynamics instead Fopt and $\alpha$ in Table 2, 3, as described in the text.

Response: Thank you for pointing this out. The Fopt and $\alpha$ were not normalised to Fopt_frac and $\alpha$_frac for all analysis, they were only normalised when the analysis was conducted for all sites. This has been clarified in the revised manuscript. In Table 2 and 3, it has also been incorporated that it was Fopt and $\alpha$ for all single site analysis, whereas it was Fopt_frac and $\alpha$_frac for all sites analysis.

Fig 3: Some points of ML-Kem are quite low (nearly 0) for MODIS GPP, while around 8 g C m-2 d-1 for EC GPP. Why?

Response: Kelma is an inundated Acacia forest located in a clay-soil depression. These differentiated values are from the beginning of the dry season, when the depression continues to have high $CO_2$ fluxes since it is still inundated, whereas, the larger area was turning dry. The EC based footprint covers this depression and in-situ GPP was thereby high, whereas the satellite based GPP covering the larger area estimated low values. This info is included in the revised manuscript.

Please unify the descriptions: $\alpha$ instead of QE, as described in the text. Clarify the labels and scales on X-axes.

Response: We have now inserted $\alpha$ into the figures. Scales has been unified on the x-axis.

(f) What is the reason that VI decreased less than 0.15 before the growing season in 2007 at NE-WaM?

Response: There are two possible reasons: 1) Uncertainty in the remote sensing data. The end of the dry season and the beginning of the rainy season is the period

of highest uncertainty in the satellite data due to aerosol and cloud contamination. This could possibly affect the VI to a low value. 2) Another possible explanation is that NE-WaM is a millet field. Agricultural practice is that before the rainy season farmers cut the shrubs in their fields. The fields are thereby cleared of vegetation before the sowing, which would decrease the VI substantially.

Please also note the supplement to this comment:
http://www.biogeosciences-discuss.net/bg-2016-414/bg-2016-414-AC2-supplement.pdf

---

## Author Response (AR1)

**Response to Editors comments:**

Non-public comments to the Author:
1. P1 L20: No need for "human induced"; "fossil fuel emissions" already indicates its relationship to humans

**Response: This has been taken care of (L20, L41; where L is line in revised manuscript).**

2. P1 L22-26: sentence structure is awkward; please use semi-colons instead of full stops; the first word in each sub-phrase should be in lowercase; insert "and" between the second and third sub-phrases

**Response: This has been taken care of (L22-L27).**

3. Balance the Abstract with respect to the importance of EVI, NDVI, and RDVI in the analysis; you do not mention EVI

**Response: We have now incorporated which vegetation indices that were used in the analysis (L25-L27).**

4. P2 L42: "are even the main biome…"; in context of the entire sentence, I am unsure what this means; please rephrase

**Response: This has been changed to: "Mean carbon dioxide (CO$_2$) uptake by terrestrial ecosystems is dominated by highly productive lands, mainly tropical forests, whereas semi-arid regions are the main biome driving its inter-annual variability (Ahlström et al., 2015; Poulter et al., 2014). Semi-arid regions even contribute to 60% of the long term trend in the global terrestrial C sink (Ahlström et al., 2015)." (L41-44)**

5. P2 L49: is "productivity" the right word? Do you mean "production"? Please make changes throughout the manuscript.

**Response: This has been taken care of throughout the manuscript.**

6. P2 L50: "under high pressure" is ambiguous; you may rewrite as "under threat"

**Response: This has been taken care of (L51).**

7. P2 L59: "Climate is thus another factor…to their vulnerability to moisture conditions"; this sentence could be made into two separate sentences. Remove the "and" and capitalize the "S" in "semi-arid region such as…"

**Response: This has been taken care of (L57-58).**

8. Many unnecessary uses of "the"; you may remove without loss of meaning

**Response: This has been taken care of throughout the manuscript.**

9. P2 L63: "defined as the efficiency to convert absorbed solar light into CO2 uptake…" can be written as "defined as the conversion efficiency of absorbed sunlight to C uptake…"

**Response: This has been taken care of (L62).**

10. P2 L74: "level" is not needed

**Response: This has been taken care of (L73).**

11. P3 L75-80 (and other places in the manuscript): Do not change verb tense midsentence

**Response: This has been taken care of throughout the manuscript.**

12. P3 L103-105: "To evaluate…"; awkward sentence, please rephrase; what is there to evaluate?

**Response: This has been changed to: "To investigate if the recently released MOD17A2H GPP (collection 6) product is better at capturing GPP for the Sahel than collection 5.1." (L102-105)**

13. P4 L136 (and other places in the manuscript): "according" should be "according to"

**Response: This has been taken care of throughout the manuscript.**

14. P5 L156 and P7 L236-237: these sentences are not needed; just indicate what was used/done, not what was needed; similar filler is used throughout the manuscript

**Response: Fillers have been removed throughout the manuscript.**

15. Never start a sentence with a symbol, a number, or an acronym. Please spell out each time when used at the start of a sentence; make changes throughout the manuscript.

**Response: This has been taken care of throughout the manuscript.**

16. Simplify hydrological and meteorological to "hydrometeorological"

**Response: This has been taken care of throughout the manuscript.**

17. P5 L173: "the fitting was insignificant (p-value < 0.05)…"; should it read "p-value > 0.05"?

**Response: This has been taken care of (L167).**

18. P5 L175: can be modified as "using a 30-day moving window with a 1-day time step"; not clear; please elaborate

**Response: This has been taken care of (L168).**

19. The manuscript needs some level of streamlining; e.g., results appearing in the Methods section (e.g., P6 L213-214) should be moved to their appropriate section; redundant material throughout the manuscript should be removed (e.g., P8 L274, P10 L357-359, and other places in the manuscript)

**Response: The result in the method section in the previous version of the manuscript has been moved to the supplementary material. Redundant material throughout the manuscript has been removed.**

20. P5 214-216: I am unsure what you mean by this statement

**Response: This section has been completely changed (L206-L214).**

21. Many statements in the manuscript are vague in nature, please be more specific (see e.g., P5 L178 and L214-215, and P10 L355 with "…to a certain level…")

**Response: We have tried to be more specific throughout the revised manuscript.**

22. P7 L247: I am unsure what is meant by "robustness"; please be specific

**Response: This has been revised throughout the manuscript.**

23. P8 L253-270: simplify detail; you may consider placing intermediate equations in a Table

**Response: We fully agree, we have simplified this section and removed some of the unnecessary intermediate equations (Section 2.4.2).**

24. P8 L273: "We used 200 iterations and different measurements sites…"; this suggests that 200 different measurement sites were used; I know this is not the case, please rephrase if the sentence is needed; i.e., "different" is vague, be specific

**Response: This has been clarified in the revised manuscript (L258-260).**

25. P8 L280: "left-out subsamples"; this was not addressed before, please introduce in the appropriate place

**Response: Within the bootstrap simulation methodology some sites were included and some were left-out. This has been clarified in the revised manuscript (L258-260).**

26. P8 L282 (and other places in the manuscript): "in situ variables" is better termed as "independent variables"; the emphasis is on the fact that the variables are independent predictors of a dependent variables opposed to the variables being measured in the field

**Response: This has been changed throughout the manuscript.**

27. A flowchart of methods and information flow would be helpful in understanding the work

**Response: It is our hope that the revised clarified manuscript makes this flowchart unnecessary. However, if this is still considered required after the review of the revised manuscript we are naturally willing to include such a flowchart.**

28. P10 L335: "works on average well…" should be "works well on average…"; similar constructions can be found throughout the manuscript, please consider changing

**Response: This has been taken care of throughout the manuscript.**

29. P11 L366-367: "…including blue-band information…"; what is the significance of this? Also, the entire sentence (L366-368) needs revising, currently awkward

**Response: We agree, this sentence has been removed.**

30. I am pleased to see that you have considered to incorporate additional maps showing regional impact on GPP and other variables; this added information should help you develop a more convincing Discussion; redundancy in the current discussion should be removed

**Response: Redundancy in the previous discussion has been removed and the additional maps are discussed in the revised manuscript.**

31. The heading of Table 3 is awkward, "statistics" do not "study"; please revise

**Response: This has been taken care of (Table 3).**

**Response to comments by Reviewer #1**

A. Summary
This paper uses data from six eddy covariance flux sites distributed across the Sahel of West Africa to examine patterns in space and time of carbon fluxes (GPP) as characterized by two key canopy-scale parameters (maximum photosynthetic uptake, called Fopt in this paper, and initial quantum yield, termed alpha). The authors also explore the relationships between the two GPP parameters and a variety of satellite vegetation indices providing (in theory at least) opportunities for spatial upscaling of the site-based results. This is an interesting paper reporting useful results.

**Response: Thank you very much, and also thank you for insightful comments that helped improving the manuscript.**

B. Main Points
1. Regional GPP estimation. It is a pity the authors didn't take the final step to evaluate GPP across the region using the fitted models. At least, we don't see a map of these estimates, only point-based comparisons with the 6 field sites. In Section 2.4.1 the authors describe a "full model" for the regression tree used to characterize fluxes and predict Fopt and alpha at the field sites. In Section 2.4.2 they continue to describe an approach to derive parameters on a pixel-by-pixel basis where not all edaphic data (e.g. soil moisture) are available. However, we don't see the results of this analysis in the form of a map or other representation. Could this be added?

**Response: We agree with the reviewer; in the previous version of the manuscript we did not include the full gridded map because the spatial up-scaling requires some very heavy computer processing. However, we have now borrowed computer power from the university, and in the revised version of the manuscript we have included a full gridded map of peak $F_{opt}$, peak α and an annual sum of GPP (L322-L326; where L is line in revised manuscript , Fig. 5).**

2. Prior work: The authors should refer to some considerable prior work that will be relevant to this analysis. See Global Change Biology 4, 523-538 (1998) and numerous HAPEX-Sahel papers in the J. Hydrology 1997 for earlier and quite detailed analysis of flux measurements in Sahelian vegetation. The GCB paper, for example, analyses Fopt and alpha as a leaf-level variable in considerable detail. Note that the canopy scale Fopt and alpha investigated here incorporate the effects of changing LAI during the season. This rather complicates the situation for this analysis, as the authors state on line 351.

**Response: Thank you for this suggestion, we agree that it was a good idea to extend the comparison of the results of our analysis to the results of previously published research. This has been incorporated into the revised manuscript (L360-374).**

3. Peak uptake rates: the field measurements at some sites seem abnormally high. The earlier data in the GCB paper references above was for a southern Sahel site with LAI likely higher than any of these sites, but with maximum Fopt of only -15-20 umol m-2 s-1.

**Response: 1) The leaf area index value of the HAPEX–Sahel West-Central fallow savanna site in (Hanan et al., 1998) is not larger than at the Dahra and Kelma sites, which are the two sites of our study with very high $F_{opt}$ and α. Peak LAI is 2.1 for Dahra and 2.7 for Kelma, so it is considerably higher than 1.2 as given in (Hanan et al., 1998). The higher LAI can thereby explain parts of the higher $F_{opt}$ estimates.**
**2) (Hiernaux et al., 2009) and (Dardel et al., 2014) showed above ground peak biomass in southwestern Niger which are comparable, and nowadays slightly lower than what is reported for the Gourma area (which in addition receives less rain).**

**Hanan et al 1997 (J Hydrology) report above ground peak biomass of 1000 and 1500 kg/ha for the grass and shrub fallow sites, which is much lower than what is reported for the Dahra site (Mbow et al., 2013), which also receives less rainfall. This is in line with a productivity gradient over these 3 sites, possibly caused by soil fertility and fallow management in southwestern Niger.**

**3) The reason for high estimates of $F_{opt}$ and α are the very high net CO2 fluxes measured by the eddy covariance systems. For the Dahra field site, we have performed a rigorous quality check of the data, please see (Tagesson et al., 2016) and we are certain that the measured values are correctly measured. Tagesson et al. (2016) have tried to explain the high net $CO_2$ flux values by that there is a combination of dense herbaceous C4 ground vegetation, high soil nutrient availability, a grazing pressure resulting in compensatory growth and fertilization effects, and the West African Monsoon bring a humid layer of surface air from the Atlantic, possibly increasing vegetation productivity for the most western part of Sahel. This info has been included in the revised manuscript (L360-374).**

4. Possible unit issues: this is an impertinent question, but looking at the massive multipliers between the author's estimates and independent estimates in Figures 2 (incoming PAR) and 3 (GPP) I couldn't help wondering if there might be some unit issues. In the case of PAR the conversion of PAR in W/m2 to umol m-2 s-1 varies somewhat based on solar angle and atmospheric conditions but is typically 4.2 umol/W. This is more than the 3.09 of the fitted slope, but is it really possible that the ERA PAR product is underestimating actual incoming PAR so consistently by a whopping 70% ! Similarly for Figure 3, if the MODIS product is in units of g/m2/day carbon and the authors have retained their data in units g/m2/day CO2 this would give an inherent slope in Figure 3 of 12/44 = 0.273. Again this doesn't entirely account for their calculated slope of 0.17, but might be worth double-checking.

**Response: Yes, we absolutely understand your concern here, and we have been looking at these conversions many times to make absolutely sure that the conversions are correctly done:**

1. **PAR values:**
**The average raw in-situ PAR = 483 μmol m-2 s-1**

**The average raw ECMWF PAR = 350503 (J m-2 summed for 3 hours)**

**To get ECMWF PAR to (W m-2): raw ECMWF PAR was divided by (60sec*60 minutes*3 hours) =>**
**Average ECMWF PAR (W m-2) =350503/(60*60*3)= 32 W m-2.**

**To convert ECMWF PAR (W m-2) to μmol m-2 we multiplied with 4.57 (Sager and McFarlane, 1997):**

**Average ECMWF PAR (μmol m-2 s-1) =32*4.57= 148 μmol m-2 s-1**

**Average in-situ PAR (μmol m-2 s-1)/ Average ECMWF PAR (μmol m-2 s-1) = 483/148 = 3.2**

**So we think that the PAR conversion is correctly done. We recently found out that the issue is related to a major error in the code of ECMWF surface PAR:**
**"The surface incident value (code 58) seems erroneously low. For example, in locations in the Celtic Sea, surface PAR is typically around 20% to 25% of the clear sky value (code 20), and about a third of in-situ measurement of surface PAR. Cause: We have shortwave bands that include 0.442-0.625 micron, 0.625-0.778 micron and 0.778-1.24 micron. PAR is coded as if it was intending to sum all of the radiation in the first of these and 0.42 of the second (to**

account for the fact that PAR is normally defined to stop at 0.7 microns. However, PAR is in fact calculated from the sum of the second band plus 0.42 of the third." (ECMWF, 2016).

This indicates that the ERA-interim surface PAR product is actually not PAR, but rather incoming red and near infrared. However, we still intend to use this data source since we relate the gridded ECMWF PAR to in-situ measured PAR and used this relationship to convert ECMWF PAR to the proper level. The relationship should be ok, even if it is relating in-situ PAR to a different part of the spectrum; the final product is still PAR at a reasonable level. The conversion of ERA interrim PAR is described in the revised manuscript (L207-214).

**2. MODIS GPP:**
An example for GPP of Agofou:

Average in-situ GPP -1.34 µmol $CO_2$ m-2 sec-1

Convert it to g $CO_2$ m-2 and s-1:

mol=44 g $CO_2$ and micro=$\mu$=$10^{-6}$

⇨ **Average in-situ GPP =0.000059 g $CO_2$ m-2 s-1**

Convert it to g $CO_2$ m-2 and 8 d-1:

days = (8*24*60*60) seconds

0.000059 g $CO_2$ m-2 s-1 * (8*24*60*60)

⇨ **Average in-situ GPP =40.7 g $CO_2$ m-2 and 8 day-1:**

Convert it to g C m-2 and 8 d-1:

g $CO_2$ = 0.27 g C

Average in-situ GPP = 40.7*0.27= 11.0 g C m-2 and 8 day-1:

Average raw MODIS GPP for Agofou: 24.1

Scaling factor: 0.0001 =>

Modis GPP (kg C m-2 and 8 day-1)=0.00241 kg C m-2 and 8 day-1

Modis GPP (g C m-2 and 8 day-1)=0.00241 *1000 = 2.41 g C m-2 and 8 day-1.

Again, we agree that this major underestimation is strange, but we believe that all conversions are correctly done.

C. Minor Points
Line 42: While it is appropriate to mention that significant inter-annual variability in global carbon cycle arises in semi-arid regions relating to rainfall variability and fire (particularly in the mesic savannas, more so than the Sahel; eg. Williams et al Carbon Balance and Management 2007), it would be an exaggeration to state that the semiarid regions are "driving long-term trends".

**Response: We agree with the reviewer that this was not a very clear sentence. But still, according (Ahlström et al., 2015) semi-arid region are driving the long term trends. We have clarified this in the revised manuscript:**

**"Vegetation growth in semi-arid regions is an important sink for fossil fuel emissions. Mean carbon dioxide ($CO_2$) uptake by terrestrial ecosystems is dominated by highly productive lands, mainly tropical forests, whereas semi-arid regions are the main biome driving its inter-annual variability (Ahlström et al., 2015; Poulter et al., 2014). Semi-arid regions even contribute to 60% of the long term trend in the global terrestrial C sink (Ahlström et al., 2015)."(L41-44)**

Line 52: "continuous cropping" is very rare in the Sahel (outside of areas with irrigation opportunities, anyway). In the drier northern regions pastoralist communities may attempt a dryland crop, but with little expectation of success. Even in the wetter southern Sahel where the crop site in this paper is located, most fields are fallowed. In the highly populated regions near the capital city of Niger, rotations have reduced, but it would be wrong to imply that "continuous cropping is practiced" widely.

**Response: Thank you for noticing this, this sentence has been removed.**

Line 107: "find evidence" is awkward here. Perhaps substitute "characterize".

**Response: Yes, we fully agree. Characterize is much better. Thank you very much.**

**Average ECMWF PAR (µmol m-2 s-1) =32*4.57= 148 µmol m-2 s-1**

**Average in-situ PAR (µmol m-2 s-1)/ Average ECMWF PAR (µmol m-2 s-1) = 483/148 = 3.2**

**So we think that the PAR conversion is correctly done. We recently found out that the issue is related to a major error in the code of ECMWF:**
**"The surface incident value (code 58) seems erroneously low. For example, in locations in the Celtic Sea, surface PAR is typically around 20% to 25% of the clear sky value (code 20), and about a third of in-situ measurement of surface PAR. Cause: We have shortwave bands that include 0.442-0.625 micron, 0.625-0.778 micron and 0.778-1.24 micron. PAR is coded as if it was intending to sum all of the radiation in the first of these and 0.42 of the second (to account for the fact that PAR is normally defined to stop at 0.7 microns. However, PAR is in fact calculated from the sum of the second band plus 0.42 of the third." (ECMWF, 2016).**

**This indicates that the ERA-interim surface PAR product is actually not PAR, but rather incoming red and near infrared. However, we still intend to use this data source since we relate the gridded ECMWF PAR to in-situ measured PAR and used this relationship to convert ECMWF PAR to the proper level. The relationship should be ok, even if it is relating in-situ PAR to a different part of the spectrum; the final product is still PAR at a reasonable level. The conversion of ERA interim PAR is described in the revised manuscript (L207-214).**

3. This paper aims to provide a model to scale up observed canopy scale GPP to regional or continental scales, using EO-based spectral vegetation indices. The readers will expect a final map of spatial distribution of GPP in semi-arid areas, and the map would make this paper more valuable.

**Response: We agree with the reviewer, in the previous version we did not include the full gridded map because the spatial up-scaling requires some very heavy computer processing. However, we have now borrowed computer power from the university, and in the revised version of the manuscript we have included a full gridded map of average peak $F_{opt}$, average peak α and an average annual sum of GPP 2001-2014 (L322-L326; Fig. 5).**

Minor comments:
Line 184: What do you mean by "air-water interface"?

**Response: We agree that the formulation was not clear. This has been corrected in the revised manuscript:**

**"The NIR radiance is reflected by the leaf cells since an absorption of these wavelengths would result in overheating of the plant whereas red radiance is absorbed by chlorophyll and its accessory pigments (Gates et al., 1965)." (L177-178)**

Table 2: Correlation between "intra-annual" dynamics

**Response: Thank you for pointing this out. This has been taken care of.**

Please unify the descriptions: use $F_{opt\ frac}$ and _frac for intra-annual dynamics instead $F_{opt}$ and α in Table 2, 3, as described in the text.

**Response: Thank you for pointing this out. The $F_{opt}$ and α were not normalised to $F_{opt\_frac}$ and $α_{\_frac}$ for all analysis, they were only normalised when the analysis was conducted for all sites. This has been clarified in the revised manuscript (L222-334). In Table 2 and 3, it has also been incorporated that it was $F_{opt}$ and α for all single site analysis, whereas it was $F_{opt\_frac}$ and $α_{\_frac}$ for all sites analysis.**

Fig 3: Some points of ML-Kem are quite low (nearly 0) for MODIS GPP, while around 8 g C m-2 d-1 for EC GPP. Why?

**Response: Kelma is an inundated Acacia forest located in a clay-soil depression. These differentiated values are from the beginning of the dry season, when the depression continues to have high $CO_2$ fluxes since it is still inundated, whereas, the larger area was turning dry. The EC based footprint covers this depression and in-situ GPP was thereby high, whereas the satellite based GPP covering the larger area estimated low values. This info is included in the revised manuscript (L275-278).**

Please unify the descriptions: α instead of QE, as described in the text. Clarify the labels and scales on X-axes.

**Response: We have now inserted α into the figures. Scales has been unified on the x-axis.**

(f) What is the reason that VI decreased less than 0.15 before the growing season in 2007 at NE-WaM?

**Response: There are two possible reasons: 1) Uncertainty in the remote sensing data. The end of the dry season and the beginning of the rainy season is the period of highest uncertainty in the satellite data due to aerosol and cloud contamination. This could possibly affect the VI to a low value. 2) Another possible explanation is that NE-WaM is a millet field. Agricultural practice is that before the rainy season farmers cut the shrubs in their fields. The fields are thereby cleared of vegetation before the sowing, which would decrease the VI substantially.**

[revised manuscript text omitted]
{\left(\rho_{\text{NIR}} - \rho_{\text{SWIR}12}\right)}{\left(\rho_{\text{NIR}} + \rho_{\text{SWIR}12}\right)} \tag{5}$$

$$\text{SIWSI}_{16} = \frac{\left(\rho_{\text{NIR}} - \rho_{\text{SWIR}16}\right)}{\left(\rho_{\text{NIR}} + \rho_{\text{SWIR}16}\right)} \tag{6}$$

where $\rho_{\text{swir}12}$ is NBAR band 5 (1230-1250 nm) and $\rho_{\text{swir}16}$ is NBAR band 6 (1628-1652 nm). As the vegetation water content increases, the reflectance in the SWIR decreases indicating that low and high SIWSI values point to sufficient water conditions and drought stress, respectively.

**2.3.3 Incoming PAR across the Sahel**

A modified version of the ERA Interim reanalysis PAR was used in the current study as an error in the code producing these PAR estimates was identified by the data distributor causing PAR values to be too low (ECMWF, 2016b). Accordingly, incoming PAR at the ground surface from ERA Interim was systematically underestimated even though it followed the pattern of PAR measured at the six Sahelian EC sites (Fig. S1 in supplementary material). In order to correct for this error, we fitted and applied an ordinary least square linear regression between in situ PAR and ERA Interim PAR (Fig. S1). The produced PAR from this relationship is at the same level as measured PAR in situ and should be at a correct level even though the original ERA Interim PAR is actually produced from the red and near infrared part of the spectrum.

Incoming PAR at the ground surface from ERA Interim followed the pattern of PAR measured at the six sites in situ closely, but it was systematically underestimated (Fig. in supplementary material2). An ordinary least square linear regression was thereby fitted between ERA Interim PAR and PAR measured in situ (PAR$_{\text{in situ}}$=3.09* PAR$_{\text{ERA interim}}$ +23.07; coefficient of determination ($R^2$)=0.93; n=37976). We therebyaThe regression line was used.

<Figure 2>

**2.4 Data analysis**

**2.4.1 Coupling temporal and spatial dynamics in photosynthetic capacity and quantum efficiency with explanatory variables**

In a first step,Tthe coupling between intra-annual dynamics in $F_{\text{opt}}$ and $\alpha$ and the vegetation indices for the different measurement sites were studied using Pearson correlation analysis. As part of the correlation analysis, we used bootstrap simulations with 200 iterations from which mean and standard deviation of the correlation coefficients were calculated (Richter et al., 2012). Relationships between intra-annual dynamics in $F_{\text{opt}}$ and $\alpha$ and the vegetation indices for all sites combined were also analysed. In the analysis for all sites, data were normalised in order to avoid influence of the spatial and inter-annual variability. T, time series of ratios of $F_{\text{opt}}$ and $\alpha$ ($F_{\text{opt\_frac}}$ and $\alpha_{\text{frac}}$) against the annual peak values ($F_{\text{opt\_peak}}$ and $\alpha_{\text{
[revised manuscript text omitted]

[Figure]

**Land cover**

| | | |
|---|---|---|
| ■ Closed evergreen lowland forest | ■ Closed grassland | ■ Irrigated croplands |
| ■ Mangrove | ■ Open grassland with sparse shrubs | ■ Sandy desert and dunes |
| ■ Mosaic Forest/Croplands | ■ Open grassland | ■ Stony desert |
| ■ Mosaic Forest/Savanna | ■ Sparse grassland | ■ Bare rock |
| ■ Deciduous woodland | ■ Swamp bushland and grassland | ■ Salt hardpans |
| ■ Deciduous shrubland with sparse trees | ■ Croplands (>50 %) | ■ Waterbodies |
| ■ Open deciduous shrubland | ■ Croplands with open woody vegetation | ■ Cities |

**Figure 1**. Land  cover classes for the Sahel and the location of the six measurement sites included in the study. The land cover classes are based on multi-sensor satellite observations (Mayaux et al., 2003). The sites are Agoufou (ML-AgG), Dahra (SN-Dah), Demokeya (SD-Dem), Kelma (ML-Kem), Wankama Fallow (NE-WaF), and Wankama Millet (NE-WaM). The thick black line is the borders of the Sahel based on the isohytes 150 and 700 mm of annual precipitation (Prince et al., 1995)

[Figure]

**Figure** 2. Photosynthetically active radiation (PAR) measured in situ against gridded ERA Interim ground surface PAR extracted for the six measurement sites (Figure 1) across the Sahel from European Centre for Medium-Range Weather Forecasts, ECMWF (2016b). The grey line is the ordinary least square linear regression.

[Figure]

[Figure]

**Figure 32.** Evaluation of the MODIS based GPP product MOD17A2H collection 6 against eddy covariance based GPP from the six measurement sites (Fig. 1) across the Sahel. The thick black line shows the one-to-one ratio, and the  grey dotted line is the fitted ordinary least square linear regression.

[Figure]

[Figure]

**Figure 4.** Dynamics in photosynthetic capacity ($F_{opt}$) and quantum efficiency ( $\alpha$) for the six measurement sites. Included is also dynamics in the vegetation indices with highest correlation to the intra-annual dynamics in $F_{opt}$ ($VI_{Fopt}$) and to quantum efficiency ($VI_{\alpha QE}$) (Table 2). The sites are a) Agoufou (ML-AgG), b) Dahra (SN-Dah), c) Demokeya (SD-Dem), d) Kelma (ML-Kem), e) Wankama Fallow (NE-WaF), and f) Wankama Millet (NE-WaM).

[Figure]

**Figure 54.** Scatter plots of annual peak values for the six measurement sites (Fig. 1) of a) photosynthetic capacity ($F_{opt\_peak}$) and b) quantum efficiency ($QE_{peak;}$ $\alpha_{peak}$) against peak values of normalized difference vegetation index ($NDVI_{peak}$) and renormalized difference vegetation index ($RDVI_{peak}$), respectively. The annual peak values were estimated by taking the annual maximum of a two week running mean.

[Figure]

**Figure 5.** Maps of a) peak values of photosynthetic capacity ($F_{opt\_peak}$) averaged for 2001-2014, b) peak values of quantum efficiency ($\alpha_{peak}$) averaged for 2001-2014, and c) annual budgets of GPP averaged for 2001-2014.

[Figure]

[Figure]

**Figure 6.** Evaluation of the modelled gross primary production (GPP) (Eq. 13) against in situ GPP from all six measurement sites across the Sahel. The thick grey line shows the one-to-one ratio, whereas the dotted thin grey line is the fitted ordinary least square linear regression.

[Figure]

[Figure]

**Figure 7.** Evaluation of the modelled gross primary production (GPP) (Eq. 13) against in situ GPP for the six sites across Sahel (Fig. 1). The thick black line shows the one-to-one ratio, whereas the dotted thin grey line is the fitted ordinary least square linear regression. The sites are a) Agoufou (ML-AgG), b) Dahra (SN-Dah), c) Demokeya (SD-Dem), d) Kelma (ML-Kem), e) Wankama Fallow (NE-WaF), and f) Wankama Millet (NE-WaM).

---

## Editor Decision (ED1)

The revision of the manuscript provides an improvement from the earlier version. However, there remains problems w.r.t. redundancy, sentence structure, etc. The manuscript could benefit from professional editing.

[revised manuscript text omitted]

$$F_{opt} = F_{opt_peak} \times F_{opt_frac} = (k_{Fopt} \times NDVI_{peak} + m_{Fopt})(n_{Fopt} \times e^{(l_{Fopt} \times RDVI_{frac})}) \tag{11}$$

$$\alpha = \alpha_{peak} \times \alpha_{frac} = (k_{\alpha} \times RDVI_{peak} + m_{\alpha})(n_{\alpha} \times e^{(l_{\alpha} \times RDVI_{frac})}) \tag{12}$$

where $k_{Fopt}$ and $k_\alpha$ are slopes and $m_{Fopt}$ and $m_\alpha$ are intercepts of the linear regressions giving $F_{opt_peak}$ and $\alpha_{peak}$, respectively; $l_{Fopt}$ and $l_\alpha$ are coefficients and $n_{Fopt}$ and $n_\alpha$ are intercepts of the exponential regressions giving $F_{opt_frac}$ and $\alpha_{frac}$, respectively. Equation 11 and 12 were inserted into Eq. 10 and GPP were thereby estimated as:

$$GPP = -(F_{opt_peak} \times F_{opt_frac}) \times (1 - e^{\left(\frac{-(\alpha_{peak} \times \alpha_{frac}) \times PAR}{F_{opt_peak} \times F_{opt_frac}}\right)}) = -((k_{Fopt} \times NDVI_{peak} + m_{Fopt})(n_{Fopt} \times e^{(l_{Fopt} \times RDVI_{frac})})) \times \left(1 - e^{\left(\frac{-(k_\alpha \times RDVI_{peak} + m_\alpha)(n_\alpha \times e^{(l_\alpha \times RDVI_{frac})}) \times PAR}{(k_{Fopt} \times NDVI_{peak} + m_{Fopt})(
[revised manuscript text omitted]

---

## Author Response (AR2)

**Response to Editors comments:**

Dear Editor

Thank you very much for your careful reading of our manuscript. We have changed the language edits you suggested and the manuscript has been proofread by a professional language editor. We hope that you are pleased with our revised manuscript.

Regarding the redundancy you mentioned in the method section. As we used bootstrap simulations both in our analysis of coupling between temporal and spatial dynamics in $F_{opt}$ and $\alpha$, and in the parameterisation and evaluation of the GPP model, we have to mention this in both instances. We have, however, removed as much redundant text as we feel is possible.

Yours sincerely,

Torbern Tagesson and co-authors

[revised manuscript text omitted]

**Land cover**

- ■ Closed evergreen lowland forest
- ■ Mangrove
- ■ Mosaic Forest/Croplands
- ■ Mosaic Forest/Savanna
- ■ Deciduous woodland
- ■ Deciduous shrubland with sparse trees
- ■ Open deciduous shrubland
- ■ Closed grassland
- ■ Open grassland with sparse shrubs
- ■ Open grassland
- ■ Sparse grassland
- ■ Swamp bushland and grassland
- ■ Croplands (>50 %)
- ■ Croplands with open woody vegetation
- ■ Irrigated croplands
- ■ Sandy desert and dunes
- ■ Stony desert
- ■ Bare rock
- ■ Salt hardpans
- ■ Waterbodies
- ■ Cities

**Figure 1**. Land cover classes for the Sahel and the location of the six measurement sites included in the study. The land cover classes are based on multi-sensor satellite observations (Mayaux et al., 2003). The sites are Agoufou (ML-AgG), Dahra (SN-Dah), Demokeya (SD-Dem), Kelma (ML-Kem), Wankama Fallow (NE-WaF)̶ and Wankama Millet (NE-WaM). The thick black line i̶s̲a̲r̲e̲ the borders of the Sahel based on the isohytes 150 and 700 mm of annual precipitation (Prince et al., 1995).

[Figure]

**Figure 2.** Evaluation of the MODIS based GPP product MOD17A2H (collection 6) against eddy covariance based GPP from the six measurement sites (Fig. 1) across the Sahel. The thick black line shows the one-to-one ratio, and the grey dotted line is the fitted ordinary least squares linear regression.

[Figure]

**Figure 3.** Dynamics in photosynthetic capacity ($F_{opt}$) and quantum efficiency ($\alpha$) for the six measurement sites. Also included are dynamics in the vegetation indices with highest correlation to the intra-annual dynamics in $F_{opt}$ ($VI_{Fopt}$) and to quantum efficiency ($VI_{\alpha}$) (Table 2). The sites are a) Agoufou (ML-AgG), b) Dahra (SN-Dah), c) Demokeya (SD-Dem), d) Kelma (ML-Kem), e) Wankama Fallow (NE-WaF) and f) Wankama Millet (NE-WaM).

[Figure]

**Figure 4.** Scatter plots of annual peak values for the six measurement sites (Fig. 1) of a) photosynthetic capacity ($F_{opt\_peak}$) and b) quantum efficiency ($\alpha_{peak}$) against peak values of normalized difference vegetation index ($NDVI_{peak}$) and renormalized difference vegetation index ($RDVI_{peak}$), respectively. The annual peak values were estimated by taking the annual maximum of a 2--week running mean.

[Figure]

**Figure 5.** Maps of a) peak values of photosynthetic capacity ($F_{opt\_peak}$) averaged for 2001-2014, b) peak values of quantum efficiency ($\alpha_{peak}$) averaged for 2001-2014, and c) annual budgets of GPP averaged for 2001-2014.

[Figure]

**Figure 6.** Evaluation of the modelled gross primary production (GPP) (Eq. 13) against in situ GPP from all six measurement sites across the Sahel. The thick grey line shows the one-to-one ratio, whereas the  thin dotted grey line is the fitted ordinary least squares linear regression.

[Figure]

**Figure 7.** Evaluation of the modelled gross primary production (GPP) (Eq. 13) against in situ GPP for the six sites across Sahel (Fig. 1). The thick black line shows the one-to-one ratio, whereas the dotted thin grey line is the fitted ordinary least squares linear regression. The sites are a) Agoufou (ML-AgG), b) Dahra (SN-Dah), c) Demokeya (SD-Dem), d) Kelma (ML-Kem), e) Wankama Fallow (NE-WaF), and f) Wankama Millet (NE-WaM).

---

## Editor Decision (ED2)

*Gross primary production*

[revised manuscript text omitted]

[a]sample size equals 11.
[b]sample size equals 9.
* significant at 0.05 level.
** significant at 0.01 level

**Table 6.** Statistics regarding the evaluation of the gross primary production (GPP) model for the six measurement sites (Fig. 1). In situ and modelled GPP are averages $\pm$ 1 standard deviation. RMSE is the root mean square error, and slope, intercept and $R^2$ are from the fitted ordinary least squares linear regression.

| Measurement site | In situ GPP ($\mu mol\ CO_2\ m^{-2}\ s^{-1}$) | Modelled GPP ($\mu mol\ CO_2\ m^{-2}\ s^{-1}$) | RMSE ($\mu mol\ CO_2\ m^{-2}\ s^{-1}$) | slope | Intercept ($\mu mol\ CO_2\ m^{-2}\ s^{-1}$) | $R^2$ |
|---|---|---|---|---|---|---|
| ML-AgG | 5.35±6.38 | 5.97±5.80 | 2.48±0.10 | 0.84±0.003 | 1.46±0.01 | 0.86±0.002 |
| SN-Dah | 9.39±10.17 | 8.87±9.67 | 3.99±1.34 | 0.88±0.002 | 0.62±0.01 | 0.85±0.001 |
| SD-Dem | 4.26±4.55 | 3.98±3.90 | 3.15±1.06 | 0.63±0.003 | 1.31±0.007 | 0.54±0.02 |
| ML-Kem | 11.16±8.02 | 10.52±9.22 | 4.35±1.23 | 1.02±0.003 | -0.82±0.03 | 0.78±0.002 |
| NE-WaF | 5.77±4.17 | 6.63±3.53 | 2.47±1.05 | 0.70±0.005 | 2.58±0.02 | 0.69±0.003 |
| NE-WaM | 3.04±1.93 | 6.35±3.47 | 4.12±0.99 | 1.31±0.004 | 2.37±0.02 | 0.53±0.003 |
| Average | 6.73±7.72 | 7.02±7.39 | 3.68±0.55 | 0.83±0.07 | 1.34±0.82 | 0.84±0.07 |

**Table 7.** The parameters for Eq. 13 that were used in the final gross primary production (GPP) model. RMSE is the root mean square error, and $R^2$ is the coefficient of determination for the regression models predicting the different variables.

| Parameter | Value | RMSE | $R^2$ |
|---|---|---|---|
| $k_{Fopt}$ | 79.6±6.3 | 5.1±1.3 | 0.89±0.05 |
| $m_{Fopt}$ | -7.3±3.2 | | |
| $l_{Fopt}$ | 3.51±0.19 | 0.15±0.02 | 0.88±0.06 |
| $n_{Fopt}$ | 0.03±0.006 | | |
| $\alpha$ | 0.16±0.02 | 0.0069±0.0021 | 0.81±0.10 |
| $m_{\alpha}$ | -0.014±0.007 | | |
| $l_{\alpha}$ | 3.75±0.27 | 0.20±0.02 | 0.80±0.10 |
| $n_{\alpha}$ | 0.02±0.007 | | |

**Figures**

[Figure]

**Land cover**

Closed evergreen lowland forest
Mangrove
Mosaic Forest/Croplands
Mosaic Forest/Savanna
Deciduous woodland
Deciduous shrubland with sparse trees
Open deciduous shrubland

Closed grassland
Open grassland with sparse shrubs
Open grassland
Sparse grassland
Swamp bushland and grassland
Croplands (>50 %)
Croplands with open woody vegetation

Irrigated croplands
Sandy desert and dunes
Stony desert
Bare rock
Salt hardpans
Waterbodies
Cities

**Figure 1**. Land cover classes for the Sahel and the location of the six measurement sites  *of this* study. The land cover classes are based on multi-sensor satellite observations (Mayaux et al., 2003). The sites are Agoufou (ML-AgG), Dahra (SN-Dah), Demokeya (SD-Dem), Kelma (ML-Kem), Wankama Fallow (NE-WaF) and Wankama Millet (NE-WaM). The thick black line  *delineates the* borders of the Sahel based on the isohytes 150 *annual* and 700 mm  (Prince et al., 1995).

*isohytes*

[Figure]

**Figure 2.** Evaluation of the MODIS based GPP product MOD17A2H (collection 6) against eddy covariance based GPP from the six measurement sites (Fig. 1) across the Sahel. The thick black line shows the one-to-one ratio and the grey dotted line is the fitted ordinary least squares linear regression.

*[handwritten annotations: "ww" , "— ✗" , "a line suggest a "linear" relationship." , "27" , circled comma mark]*

[Figure]

*Time series of*

**Figure 3.**  photosynthetic capacity ($F_{opt}$) and quantum efficiency ($\alpha$) for the six measurement sites. Also included are  vegetation indices with highest correlation  *with* $F_{opt}$ ($VI_{Fopt}$) and  quantum efficiency ($VI_\alpha$) (Table 2). The sites are a) Agoufou (ML-AgG), b) Dahra (SN-Dah), c) Demokeya (SD-Dem), d) Kelma (ML-Kem), e) Wankama Fallow (NE-WaF) and f) Wankama Millet (NE-WaM).

*timeseries of the*

[Figure]

**Figure 4.** Scatter plots of annual peak values for the six measurement sites (Fig. 1) of a) photosynthetic capacity ($F_{opt\_peak}$) and b) quantum efficiency ($\alpha_{peak}$) against peak values of normalized difference vegetation index ($NDVI_{peak}$) and renormalized difference vegetation index ($RDVI_{peak}$), respectively. The annual peak values were estimated by taking the annual maximum of a 2-week running mean.

[Figure]

**Figure 5.** Maps of a) peak values of photosynthetic capacity ($F_{opt\_peak}$) averaged for 2001-2014, b) peak values of quantum efficiency ($\alpha_{peak}$) averaged for 2001-2014, and c) annual budgets of GPP averaged for 2001-2014.

[Figure]

**Figure 6.** Evaluation of the modelled gross primary production (GPP) (Eq. 13) against in situ GPP from all six measurement sites across the Sahel. The thick grey line shows the one-to-one ratio, whereas the thin dotted grey line is the fitted ordinary least squares  regression.

[Figure]

**Figure 7.** Evaluation of the modelled gross primary production (GPP) (Eq. 13) against in situ GPP for the six sites
 (Fig. 1). The thick black line shows the one-to-one ratio, whereas the dotted thin grey line is the fitted ordinary least

*no need to tell us that the sites are in the Sahel.*

squares linear regression. The sites are a) Agoufou (ML-AgG), b) Dahra (SN-Dah), c) Demokeya (SD-Dem), d) Kelma (ML-Kem), e) Wankama Fallow (NE-WaF) and f) Wankama Millet (NE-WaM).